

**Climate engineering and the ocean: effects on biogeochemistry and primary production**
Siv K. Lauvset[1], Jerry Tjiputra[1], Helene Muri[2],
[1]Uni Research Climate, Bjerknes Center for Climate Research, Jahnebakken 5, Bergen,
Norway
[2]University of Oslo, Department of Geosciences, Section for Meteorology and Oceanography,
Oslo, Norway
**ABSTRACT**
Here we use an Earth System Model with interactive biogeochemistry to project future ocean
biogeochemistry impacts from large-scale deployment of three different radiation
management (RM) climate engineering (also known as geoengineering) methods:
stratospheric aerosol injection (SAI), marine sky brightening (MSB), and cirrus cloud
thinning (CCT). We apply RM such that the change in radiative forcing in the RCP8.5
emission scenario is reduced to the change in radiative forcing in the RCP4.5 scenario. The
resulting global mean sea surface temperatures in the RM experiments are comparable to
those in RCP4.5, but there are regional differences. The forcing from MSB, for example, is
applied over the oceans, so the cooling of the ocean is in some regions stronger for this
method of RM than for the others. Changes in ocean primary production are much more
variable, but SAI and MSB give a global decrease comparable to RCP4.5 (~6% in 2100
relative to 1971-2000), while CCT give a much smaller global decrease of ~3%. The spatially
inhomogeneous changes in ocean primary production are partly linked to how the different
RM methods affect the drivers of primary production (incoming radiation, temperature,
availability of nutrients, and phytoplankton) in the model. The results of this work
underscores the complexity of climate impacts on primary production, and highlights that



changes are driven by an integrated effect of multiple environmental drivers, which all change
in different ways. These results stress the uncertain changes to ocean productivity in the
future and advocates caution at any deliberate attempt for large-scale perturbation of the Earth
system.
**1 INTRODUCTION**

Human emissions of carbon dioxide to the atmosphere is unequivocally causing global

warming and climate change (IPCC, 2013). At the 21st United Nations Framework
Convention on Climate Change (UNFCCC) Conference of the Parties it was agreed to limit
the increase in global mean temperatures to 2°C above pre-industrial levels and to pursue
efforts to remain below 1.5°C. Reaching this goal will not be possible without radical social
transformation. Solar radiation management (SRM) has been suggested as both a method of
offsetting global warming and to reduce risks associated with climate change, substituting
some degree of mitigation (Teller et al., 2003, Bickel and Lane, 2009), or to buy time to
reduce emissions (Wigley, 2006). Reducing the otherwise large anthropogenic-induced
changes in the marine ecosystem drivers (*e.g.*, temperature, oxygen, and primary production)
could also be beneficial for vulnerable organisms that need more time to migrate or adapt
(Henson et al., 2017).  SRM is the idea to increase the amount of solar radiation reflected by
Earth in order to offset changes in the radiation budget due to the increased greenhouse effect
from anthropogenic emissions, *i.e.* a form of climate engineering – or geoengineering.

Here we have performed model experiments with stratospheric sulfur aerosol

injections (Crutzen, 2006; Weisenstein et al., 2015), and marine sky brightening (Latham,
1990), and cirrus cloud thinning (Mitchell and Finnegan, 2009). Stratospheric aerosol
injections (SAI) would involve creating a layer of reflective particles in the stratosphere to
reduce the amount of solar radiation reaching the surface. The most widely discussed



approach to SAI is to release a gaseous sulfate precursor, like $SO_2$, which would oxidize to
form sulfuric acid and then condensate to reflective aerosol particles. Marine sky brightening
(MSB) aims to reflect the incoming solar radiation at lower levels in the atmosphere. Here,
the idea is to spray naturally occurring sea salt particles into low-lying stratiform clouds over
the tropical oceans to increase the available cloud condensation nuclei, thus increasing the
concentration of smaller cloud droplet and increase the reflectivity of the clouds (Latham,
1990). The sea salt aerosols are reflective in themselves (*e.g.*, Ma et al., 2008), adding to the
cooling potential of the method. Cirrus cloud thinning (CCT) on the other hand, aims to
increase the amount of outgoing longwave radiation at the top of the atmosphere. This is
envisioned done by depleting the longwave trapping in high ice clouds by seeding them with
highly potent ice nuclei (*e.g.*, Mitchell and Finnegan, 2009; Storelvmo et al., 2013). In the
absence of naturally occurring ice nuclei, the seeded material would facilitate freezing at
lower supersaturations, enabling the growth of fewer and larger ice crystals. These would
eventually grow so large that they sediment out of the upper troposphere reducing the lifetime
and optical thickness of the cirrus clouds leading to a cooling effect. Together these three
methods are referred to as Radiation Management (RM).
As pointed out by Irvine et al. (2016) there are several gaps in the research on the
impact of RM on both global climate and the global environment considering only a few
modelling studies to date systematically compare multiple RM methods. Aswathy et al.
(2015) and Niemeier et al. (2013) compared stratospheric sulfur aerosol injections to
brightening of marine clouds in terms of the hydrological cycle and extremes in temperatures
and precipitation. Crook et al. (2015) compared the three methods used in this study, but
restricted the study to temperatures and precipitation. This study focuses on the impact on the
ocean carbon cycle, which has several potential climate feedbacks (Friedlingstein et al.,



2006), and in particular on ocean primary production, which is known to be temporally and
spatially complex.

The effect RM has on the ocean carbon cycle and ocean productivity has been studied

previously, but limited to the use of simple one-dimensional models (Hardman-Mountford et
al., 2013) or with global models but focusing on a single method of RM (Partanen et al.,
2016; Tjiputra et al., 2015, Matthews et al., 2009). Due to the many uncertainties and open
questions associated with RM impacts, a systematic comparative approach is necessary. The
three different methods of RM used in this study are likely to have different effects on both
the climate and the ocean due to the differences in the type of forcing being applied. An
aspect of RM is that it may allow for continued $CO_2$ emissions in the future without the
accompanied temperature increases and that it does not directly affect the atmospheric $CO_2$
concentrations. Ocean acidification, a direct consequence of increased $CO_2$ concentrations in
the atmosphere, would therefore continue with RM, unless paired with mitigation and carbon
dioxide removal.

This manuscript is the first to evaluate and compare the effect and impact of multiple

RM techniques on ocean biogeochemistry using a fully coupled state-of-the-art Earth system
model, and furthermore extends previous studies by looking into impacts introduced by three
different large-scale RM deployment scenarios both during and after deployment periods. It is
also the first study to assess the impacts of cirrus cloud thinning on ocean biogeochemistry.
Our focuses are on impacts on SST, oxygen, pH, and primary production, which are the four
climate drivers identified by the Intergovernmental Panel on Climate Change (IPCC),
significantly affecting marine ecosystem structure and functioning. In a wider perspective,
ocean primary production is often used as an indicator for marine food availability, such as
fisheries, so furthering our understanding has direct societal implications and a strong
connection to the United Nations Sustainable Development Goals.





The model and experiments are described in detail in Section 2, the impacts on ocean
temperature, oxygen content, inorganic carbon, and primary production are presented and
discussed in Section 3, while Section 4 summarizes and concludes the study.

**2 METHODS**
**2.1 Model description**
Three RM methods are simulated using the Norwegian Earth System Model
(NorESM1-ME; Bentsen et al., 2013). The NorESM1-ME is a fully coupled climate-carbon
cycle model, which has contributed to the fifth assessment of the IPCC and participated in
numerous Coupled model intercomparison project phase 5 (CMIP5) analyses. For a full
description of the physical and carbon cycle components of the model, the readers are referred
to Bentsen et al. (2013) and Tjiputra et al. (2013), respectively. Here, we only briefly describe
some key processes in the ocean carbon cycle that are relevant for this study.
The ocean carbon cycle component of the NorESM1-ME originates from the Hamburg
Oceanic Carbon Cycle Model (HAMOCC; Maier-Reimer et al., 2005). In the upper ocean,
the lower trophic ecosystem is simulated using an NPZD-type (Nutrient-Phytoplankton-
Zooplankton-Detritus) module. The primary production depends on phytoplankton growth
and nutrient availability within the euphotic layer (for some of our calculations assumed to be
100 m). In addition to multi-nutrient limitation, the phytoplankton growth is light- and
temperature-dependent. The net primary production (NPP) in NorESM1-ME is parameterized
using the equations of Six and Maier-Reimer (1996) (Equation 1).
$NPP = r(T,L) * \frac{N}{N+No} * P$                               Equation 1
where  $r(T,L) = \frac{f(L)*f(T)}{\sqrt{(f(L)^2+f(T)^2)}}$                               Equation 2





$N$ is the concentration of the limiting nutrient (either phosphate, nitrate or dissolved iron), $f(L)$
is the function determining light-dependency, $f(T)$ is the function for temperature-dependency,
and $P$ is the phytoplankton concentration. Both $f(L)$ and $f(T)$ are defined in Six and Maier-
Reimer (1996).

In addition to the growth through NPP, the phytoplankton has several sink terms due

to mortality, exudation, and zooplankton grazing. All nutrients, plankton, and dissolved
biogeochemical tracers are prognostically advected by the ocean circulation. The model
adopts a generic bulk phytoplankton and zooplankton compartments. The detritus is divided
into organic and inorganic materials: particulate organic carbon, biogenic opal, and
calcium carbonate. Organic carbon, once exported out of the euphotic layer, is remineralized
at depth – a process that consumes oxygen in the ocean interior.  Non-remineralized particles
reaching the seafloor undergo chemical reactions with sediment pore water, bioturbation, and
vertical advection within the sediment module. The model calculates air-sea $CO_2$ fluxes as a
function of seawater solubility, gas transfer rate, and the gradient of the gas partial pressure
($pCO_2$) between atmosphere and ocean surface, following Wanninkhof (1992). Prognostic
surface ocean $pCO_2$ is computed using inorganic seawater carbon chemistry formulation
following the Ocean Carbon-cycle Model Intercomparison Project (OCMIP).

In this study, we make use of ocean primary production calculations made both online

by NorESM1-ME and offline, using the monthly averaged output from the model. The offline
calculations make use of the same set of equations as the online calculation, but (i) the
average value over the top 100 m is used for $N$, $T$, and $P$ alike; (ii) $L$ is attenuated to a
constant depth of 50 m; (iii) the monthly mean is used for $N$, $T$, $L$, and $P$. The calculation
allows us to decompose and identify the dominant drivers for the simulated changes. The
decomposition is done by choosing to keep all but one parameter $x$ constant at a time to
quantify the contribution of parameter $x$ to the total change. Table 1 describes how this was



done. The parameters being kept constant are kept at the long-term (80 year) monthly mean,
as calculated from the pre-industrial model experiment (with constant atmospheric $CO_2$
concentrations).

**2.2 Experiment setup**

SAI, MSB, and CCT were applied individually to the RCP8.5 (Representative

Concentration Pathway) future scenario (Table 2). The target of the simulations were to
reduce the global mean top of the atmosphere (TOA) radiative flux imbalance of RCP8.5
down to RCP4.5. In each experiment, the forcing is applied over years 2020 to 2100. To study
the termination effect, the simulations are continued for another 50 years following the
cessation of each RM method.

Here, the SAI, MSB, and CCT experiments are analyzed and compared to the RCP4.5

and RCP8.5 scenarios (Riahi et al., 2011; Thomson et al., 2011) (Table 2). All simulations are
run with interactive biogeochemistry and use prescribed anthropogenic $CO_2$ emissions. The
atmospheric $CO_2$ concentrations are therefore prognostically simulated accounting for land-
air and sea-air $CO_2$ fluxes.

The SAI was implemented by prescribing a global layer of sulfate aerosols in the

stratosphere, and the optical properties were taken from the ECHAM dataset described in
Tilmes et al. (2015). The injection strength was scaled up to 20 TgS in year 2100. The MSB
follows the method of Alterskjaer et al. (2013), where the emissions on accumulation mode
sea salt was increased over the oceans. Here we choose to apply this to a latitude band of
±45°. The tropospheric aerosol scheme is fully prognostic, thus allowing the full interactive
cycle with clouds and radiation. As for the CCT, we adopt the approach of Muri et al. (2014),
where the terminal velocity of ice crystals at typical cirrus forming temperatures of colder
than -38 °C is increased. The maximum effective radiative forcing was found to be limited at





about -3.8 W m$^{-2}$ for CCT, resulting in a somewhat higher top of the atmosphere (TOA)
radiative flux imbalance in this simulation at 2100.

**3 RESULTS AND DISCUSSION**
**3.1 Global changes in ocean temperature and oxygen concentration**
Relative to the 1971-2000 historical period, the ocean oxygen content in the 200-600
m depth interval is projected to decrease by ~6% globally in 2100 in RCP8.5 (Figure 1a). In
RCP4.5 on the other hand, the inventory of oxygen in the 200-600 m interval shows only a
minor decrease of 2% by 2100 (Figure 1a). This difference stems partly from lower oxygen
solubility as the ocean warms and partly from changes in ocean stratification and circulation
(not shown). When applying RM to RCP8.5, the oxygen concentration in this depth interval
follows the RCP4.5 development closely for all three RM methods (ranging from 2-2.6%
decrease in 2100 compared to the 1971-2100 average). There are, however, differences
between the methods, with SAI yielding slightly larger decreases after 2060 (Figure 1a). After
termination of RM, the rate of oxygen reduction accelerates rapidly for the first ten years,
before stabilizing at a new rate of decrease of similar magnitude to that in RCP8.5. The
projected oxygen reductions do not drop as low as in RCP8.5 after termination of the RM
during our simulation period, but had the simulations been continued for some further
decades, the oxygen levels would most likely have converged to the RCP8.5 levels. In 2150,
RCP8.5 shows a global mean oxygen decrease globally of 9.5%, while the simulations with
terminated RM show a global mean oxygen decrease of 8-8.5% (Figure 1a).
In RCP8.5, the global mean sea surface temperatures (SST) are projected to increase
by ~2.5 °C by 2100 relative to 2010 (Figure 1b), and ~3 °C relative to the 1971-2000 average.
With RM, the changes in SST are kept similar to RCP4.5, with an increase ranging from 0.8



to 1.1°C over the time period between 2020 (start of RM deployment) and 2100. After
termination, there is a very rapid SST increase in the subsequent decade before the SST
increases more gradually towards that in RCP8.5. Similar to the development in oxygen
content, the absolute change in SST in the model runs with terminated RM is still smaller than
the absolute change in RCP8.5 (Figure 1b) in 2150. This is mainly due to the slow response
time of the ocean, so the SST would eventually converge had the simulations been carried out
for a longer period of time after termination. It should be noted that all methods of RM used
in this study have been designed to produce the global mean radiative forcing at the end of the
century that is equivalent to the difference in the anthropogenic radiative forcing between
RCP4.5 and RCP8.5, *i.e.* 4 W m$^{-2}$. This means that the globally averaged sea surface
temperature changes, and changes in large-scale physical variables such as oxygen, are
expected to be close to those in RCP4.5. The results presented here imply that applying RM
does not prevent the long-term impacts of climate change, but would on average delay them.
In the case of oxygen concentrations in the 200-600 m depth interval the changes incurred in
RCP4.5, as well as when the three different methods of RM are applied, are mostly not
significantly different (*i.e.* they are smaller than one standard deviation) from the 1971-2000
average (Figure 2). There are a few exceptions where the oxygen changes are significant.
These regions, however, highlight how differently the RM methods affect the ocean.

The spatial absolute change in SST in 2071-2100 relative to 1971-2000 is shown in

Figure 3b for RCP8.5 and Figure 3c for RCP4.5. The changes are significantly smaller in
RCP4.5, but the spatial variations are the same in RCP8.5 and RCP4.5. When applying RM,
the changes in SST are everywhere smaller than in RCP8.5 at the end of the century. As for
thermocline oxygen, the spatial patterns are altered in some regions, as seen in the zonally
averaged temperature changes (Figure 3a). The SAI method yields the temperature change
most similar to that in RCP4.5, which is also mirrored in the near surface air temperatures





(Muri et al., in prep). MSB yields the SST changes that are most different compared to
RCP4.5. For this method there is a strong bimodal pattern in the SST changes in the North
Pacific (Figure 3e), which is also seen in oxygen (Figure 2e). The tropical and subtropical
changes in SST with MSB are linked to an enhancement of the Pacific Walker cell, which is
induced when MSB is applied (Alterskjær et al., 2013; Ahlm et al, 2017).
Regardless of the RM method, some regions, in particular the northwestern Pacific,
will still experience levels of warming (cooling) and oxygen loss (gain) exceeding those in
RCP4.5. With SAI, the North American west coast, an important region for aquaculture, will,
for example, experience enhanced deoxygenation, which is not projected to happen in
RCP4.5. The large spatial heterogeneity in how RM affects ocean temperatures and oxygen
concentrations highlights that RM can possibly lead to new and detrimental conditions
regionally even if beneficial in the global mean.

**234   3.2 Global changes in the inorganic ocean carbon cycle**

The atmospheric $CO_2$ concentration continue to rise in all experiments in which RM is
applied at the same rate as in RCP8.5 (Figure 4a), given no simultaneous mitigation efforts in
these cases. The atmospheric $CO_2$ concentration in 2100 in RCP8.5 is 1109 ppm and in 2150
it is 1651 ppm. In 2100 there is a minor reduction in $CO_2$ concentrations when RM is applied
of 13 -21 ppm compared to RCP8.5, depending on method. MSB gives the largest decrease in
atmospheric $CO_2$. The termination of RM does not significantly affect the atmospheric $CO_2$
evolution and in 2150 there is a marginal reduction of -15 to -26 ppm depending on method,
again with MSB giving the largest reduction. The reductions in atmospheric $CO_2$
concentrations when applying RM are due to the decreasing ocean temperatures leading to
larger air-sea flux of $CO_2$ (Figure 4b). Note that the land carbon sinks also increase slightly



when RM is applied (Tjiputra et al., 2016). The lower $CO_2$ concentration with MSB is due to
the forcing from MSB being applied over the oceans, and the cooling of the ocean in many
regions thus being stronger for this method of RM (Figure 3).

While RM leads to a small increase global mean oceanic $CO_2$ uptake from the

atmosphere, due to increased solubility, the difference introduced by each method is not
outside of the interannual variability of RCP8.5 up to 2075. By 2100, the different RM
methods give an additional $CO_2$ uptake of ~0.5 PgC yr$^{-1}$. After termination, the uptake
anomaly quickly drops and returns to the same level as RCP8.5 within only two years. Future
surface ocean pH is forced by the increasing atmospheric $CO_2$ concentrations, which drive the
uptake of $CO_2$ in the surface ocean. Thus RM could possibly worsen future ocean
acidification, unless atmospheric $CO_2$ concentrations are dealt with. However, given the small
changes in both atmospheric concentrations and ocean uptake stemming from RM, the surface
pH is not greatly affected by RM (Figure 4c). Hence, termination does not considerably affect
the pH decrease on the surface ocean.

Anthropogenic changes in the ocean inorganic carbon content comes from the top

down, so it takes a long time for these changes to be observable in the deep ocean. Therefore,
the globally averaged deep ocean (>2000 m) pH changes by only 0.06 pH units between 2010
and 2150 in RCP8.5 (Figure 4d). The only region where pH changes significantly in the deep
ocean is the North Atlantic north of 30˚N, where the strong overturning circulation brings
anthropogenic carbon to great depths in a relatively short timeframe. Here there is a
significant decrease in deep ocean pH between 2010 and 2150 in RCP8.5, as well as the three
RM cases (Figure 4e). In RCP8.5, the pH is projected to decrease by ~0.2 pH unit in 2100.
RM leads to an additional acidification of 0.02-0.045 (depending on the method of RM) in the
deep North Atlantic Ocean, which is large enough to marginally, but not significantly, affect
the global average (Figure 4d). A similar result was found by Tjiputra et al. (2015). After



270 termination of RM, the pH keeps decreasing – now at a rate comparable to RCP8.5. This

271 change in rate of decrease after termination happens within ~10 years, indicating that the

272 changes in the inorganic carbon cycle are very quick in the North Atlantic. Both the rapid

273 decrease of deep ocean pH in this region and the rapid recovery towards RCP8.5 development

274 after termination of RM, are likely linked to changes in the Atlantic Meridional Overturning

275 Circulation due to climate change and RM (not shown, see Muri et al., in prep.). While the

276 global mean pH below 2000m in RM experiments rebound to that of the RCP8.5, this is not

277 the case for the North Atlantic. In the latter, all RM methods lead to and remain at lower pH

278 than the RCP8.5 by 2150. It is likely that the deep pH in the North Atlantic would recover to

279 that in RCP8.5 had the simulations been continued for another few decades, but we have no

280 way of analyzing how long that would take.

## 3.3 Global changes in ocean primary production

283   The direct effects of RM on surface shortwave radiation and temperature directly

284 affect photosynthesis through the light and temperature dependence of the phytoplankton

285 growth rate. The ocean productivity, and by extension ocean biological carbon pump, is thus

286 indirectly affected by RM. There is a lot of interannual variability in the primary production

287 changes hence Figure 5 shows the 5-year running averages of relative changes to the 1971-

288 2000 average. In RCP8.5, there is a decrease of ~10% by 2100 (Figure 5), which is within the

289 range of the decrease projected by CMIP5 models of -8.6±7.9% (Bopp et al., 2013) and

290 mainly due to the overall warming leading to a more stratified ocean where there are less

291 nutrients available in the euphotic zone. All RM methods also exhibit decreases in ocean

292 primary productivity, but these are all smaller than those in RCP8.5. The shortwave-based

293 methods, *i.e.*, SAI and MSB, which reduce the amount of downward solar radiation at the

294 surface, have the largest decreases (~6% in 2100) of the RM methods, which is more of a



decrease than in RCP4.5. The longwave-based CCT method, however, yields only a minor
decrease of ~3% in 2100, *i.e.* less than in RCP4.5. As the cirrus clouds are thinned or
removed, more sunlight reaches the surface ocean, thus promoting and increasing primary
above the RCP4.5 levels. The divergence between methods is particularly strong in the period
2070-2100, as the radiative forcing by RM approaches -4 $Wm^{-2}$. After termination, it takes
less than five years for the development of ocean primary production to return to RCP8.5
levels again.

On average there are some interesting spatial features in how primary production

changes. Figure 6a shows the zonally averaged difference between 2071-2100 and 1971-
2000. In the Northern Hemisphere, primary production decreases everywhere, and decreases
less in RCP4.5 and with RM than in RCP8.5. In the Southern Hemisphere, on the other hand,
the changes in primary production are much more spatially variable, and the response to the
different methods of RM is more variable. Between the Equator and 40°S there is a reduction
in primary production in 2071-2100 relative to 1971-2000, while south of 40° there is
generally an increase (except in a narrow band at 60°S). In the Southern Hemisphere the
impact of CCT is quite different from the impact of SAI and MSB. This is probably due to the
change in radiative balance, which is much stronger for CCT in the southern high latitudes
than for the other methods (not shown, see Muri et al., in prep.). Because of the large spatial
and inter-annual variability, the changes incurred to ocean primary production in the future
are frequently not significantly different (*i.e.* the absolute change is smaller than one standard
deviation) from the 1971-2000 average (Figure 6b-f). This means that when RM is applied,
the ocean primary production does not change in most of the ocean. However, it is clear that
the changes in primary production in 2071-2100 relative to 1971-2000 are smaller in RCP4.5
than in RCP8.5 (Figures 6b and 6c), and that the spatial variations in all experiments mainly
come from the nutrient availability (not shown), which is furthermore dependent on ocean



stratification. There are also some regions of significant change in ocean primary production,
which are discussed further in Section 3.5.

**3.4 Drivers of global changes in ocean primary production**

To further evaluate how RM affects ocean primary production, we have made offline
calculations using Equation 1 and the monthly mean model output of nitrate, phosphate, iron,
and phytoplankton concentration, temperature, and shortwave radiation input at the surface, as
described in Section 2. For the top 100 m of the ocean, only CCT significantly changes
primary production compared to RCP8.5. In fact, CCT results in an increased productivity by
2100 (Figure 7a), which is linked to the increase in the incoming shortwave solar radiation in
some regions, since the shortwave reflection from ice clouds is reduced. After termination of
CCT, the primary production drops to the same level as RCP8.5 within two years. The
RCP4.5 scenario yields little change by 2100. The fact that CCT shows a significant global
increase in ocean primary production relative to RCP8.5 and even a positive change at the end
of the century is a very interesting result of this study. It suggests that when considering the
global ocean primary production changes alone, implementation of CCT may offer the least
negative impact of the three tested methods. The side effect, however, is that once terminated,
CCT method could lead to most drastic change in primary production over very short period.
Warmer temperatures increase growth rates.  Thus primary production increases when
only temperature is allowed to change in the offline calculation, as temperature increases in
all scenarios considered here (Figure 7b). All methods of RM yield an increase in primary
production of ~1% from 2020 to 2100, comparable to RCP4.5, in this calculation. This is
consistent with SST being comparable between RCP4.5 and RM (Figure 1b). After
termination, the temperature-induced primary production increases rapidly for the first five



years before stabilizing with the same rate of change as that in RCP8.5. Just like SST (Figure
1b), the absolute change in primary production does not recover to the quite the same absolute
level as that in RCP8.5, but all simulations show an increase in primary production of ~3% by

2150.

Reduced shortwave radiation at the surface lead to decreased primary production. In

RCP4.5 and RCP8.5, light constraints do not change much, hence the primary production also
does not considerably change when only shortwave radiation is allowed to vary in the offline
calculation (Figure 7c). Both SAI and MSB decrease the amount of global mean direct
shortwave radiation at the surface, however, which negatively affect the phytoplankton
growth rate and primary production in the ocean (Figure 7c). The result of allowing only
shortwave radiation to vary is therefore a decrease in primary production of ~2% by 2100 for
SAI and MSB (Figure 7c). When reducing the optical thickness and the lifetime of the cirrus
clouds in the model, the shortwave reflection by these clouds is reduced, allowing more
shortwave radiation to reach the surface. CCT thus results in an increase in primary
production of ~2% by 2100 (Figure 7c). It is this increase in available shortwave radiation
that causes the majority of the increase in ocean productivity with CCT, with some
contribution from the elevated temperatures (Figure 7b). Within two years of the termination
of RM, the simulated primary production has completely returned to the baseline conditions.

Inorganic nutrients are also important limiting factors, especially in the low latitude

regions. Given the formulation of Equation 1, we use phytoplankton concentration as a proxy
for nutrient availability when calculating primary production. Note though, that the
relationship between nutrients and phytoplankton is not exactly one to one because
phytoplankton are also grazed by zooplankton in the model. However, temporal changes in
phytoplankton concentration give a strong indication of how the stratification limits access to
nutrients in the surface ocean. Figure 7d shows that phytoplankton is the dominant factor



determining changes in ocean primary production, except when CCT is applied. When only
phytoplankton concentration is allowed to vary temporally in the offline calculation there is a
decrease of ~8% by 2100 in RCP8.5. The SAI and MSB methods of RM also exhibit a change
in primary production, but the change of ~5% is less than that in RCP8.5. With CCT there is
no significant change in primary production by 2100. After termination, the phytoplankton-
driven change of ocean productivity decreases rapidly and after 4-5 years it continues
changing at a rate comparable to that in RCP8.5, reaching a global mean reduction of greater
than -10% in 2150.

### 3.5 Regional changes in ocean primary production

As seen in Figure 6, the projected changes in ocean primary production exhibit large
spatial variation. Applying RM does not change the large-scale spatial heterogeneity, but
rather works to enhance or weaken the change magnitude (Figure 6). These regional
differences are important since regional changes are much more important than global
changes when determining the impact changes in ocean primary production has on human
food security (Mora et al., 2013). For a more detailed analysis, five regions have been
identified and analyzed. These regions are chosen based on:
(i)     a significant change in primary production in RCP8.5 in years 2071-2100 relative to

1971-2000;

(ii)     the sign of the change in ocean primary production projected by NorESM1-ME being

consistent with that of the CMIP5 models ensemble (Bopp et al., 2013; Mora et al.,

2013);

(iii)     the impact the different methods of RM has on this increase or decrease in the online

simulations; and





(iv)    their relative importance for fish catches, as identified in Zeller et al. (2016).
The regions are outlined in black in Figure 6b, and labeled the Equatorial Pacific,
Equatorial Atlantic, Southern Atlantic, Indian Ocean, and Sea of Okhotsk in Figure 8. In
RCP8.5, the Sea of Okhotsk and Southern Atlantic exhibit a significant increase in primary
production in 2071-2100 relatively to 1971-2000, while the Equatorial Pacific, Indian Ocean,
and Equatorial Atlantic show a significant weakening (Figure 8). Given the lack of
complexity and lack of higher trophic level organisms in the NorESM1-ME, we are unable to
directly link changes in primary production to impacts on the higher tropic levels in this
study. But given the changes in Arctic biodiversity observed today due to temperature
changes (*e.g.* Bucholz et al., 2012; Fossheim et al., 2015), respective changes in migration
pattern would be likely to happen with RM. It cannot be assumed from our results that
increased primary production will lead to increased fish stocks and thus potential for higher
fish catches, because the driving factors leading to higher primary production (*i.e.*
temperature, light availability, and stratification) could also lead to biodiversity changes.
Higher primary production does lead to more food for higher trophic level organisms,
therefore a significant decrease in regional primary production is likely to decreases higher
tropic organisms due to less food availability in those regions. Based on the model
projections, it is possible that there will be less fish catches in the Indian Ocean and
Equatorial Atlantic in the future than today. The different methods of RM also lead to
different effects on ocean primary production (Figure 6 and 8), and in the Equatorial Atlantic
and in the shaded regions where there is no significant changes, do all three methods give
changes in primary production comparable to those in RCP4.5.
In the Equatorial Pacific RCP8.5 leads to a decrease in ocean primary production of
21% in 2071-2100 relative to 1971-2000, driven by changes in phytoplankton concentration
(our proxy for circulation changes). Changes in circulation dominates the change of 12%





incurred in RCP4.5 too. This region is today a very productive fishery area (Zeller et al.,
2016), so a significant decrease in primary production could have adverse effects on fish
catches. It is therefore noteworthy that all RM methods yield primary production changes
only marginally smaller than those in RCP8.5, and not nearly as small as those in RCP4.5.
Radiation changes become more important in driving changes with RM, which is consistent
with changes in cloud fraction (not shown, see Muri et al., in prep.). With CCT the radiation
changes yield an increase in primary production of 5% indicating that this is one of the
regions that drive the global mean increase in primary production with CCT (Figure 7a). After
termination, the change in primary production is comparable to that in RCP8.5 in all
experiments, and the warming incur a small increase in primary production of ~2%.

The Southern Atlantic has the largest changes in 2071-2100 relative to 1971-2000

where RCP8.5 results in an increase in ocean primary production of 39% and RCP4.5 leads to
an increase of 25%. SAI leads to changes in primary production comparable to that in
RCP8.5, while MSB and CCT yielding changes more in line with RCP4.5. For all
experiments changes in phytoplankton concentration is the dominant factor indicating that
changes in circulation will be substantial here. Changes in temperature contribute ~5% to the
total change which is consistent with a significant warming in all experiments (Figure 3). This
alleviates the temperature limitation of phytoplankton growth, which is consistent with the
other CMIP5 models (Bopp et al., 2013). After termination, the increase continues in the
Southern Atlantic, and in 2121-2150 the changes in primary production are 50-60% higher
than in 1971-2000 in all experiments.

In the Sea of Okhotsk changes in temperature yield changes in primary production

comparable with that in RCP4.5 (13%), which is marginally smaller than that in RCP8.5
(18%). SAI and MSB both yield changes comparable to that in RCP4.5, while CCT, on the
other hand, is comparable to RCP8.5. In all experiments, temperature changes are an



important driver of the overall increases in primary production, which is consistent with the
strong warming in this region (Figure 3). After termination, all experiments yield comparable
increases in primary production, and the temperature changes have the largest contribution to
the overall increase, which is consistent with strong warming when RM is terminated.

In the Equatorial Atlantic there is a reduction of ocean primary production in RCP8.5

of 19% in 2071-2100 relative to 1971-2000. Changes in phytoplankton concentration
dominate this change, with a minor contribution of <5% from radiation changes. All methods
of RM yield changes in ocean primary production more in line with that in RCP4.5 (11%), but
changes in radiation are more important with SAI and MSB. After termination, all
experiments result in the same decrease in ocean primary production of 25%.

In the Indian Ocean there is also a reduction of ocean primary production in RCP8.5.

Here the total change in 2071-2100 is 21%, but unlike in any other regions the temperature
induced changes lead to only a small increase of 1-2% in all experiments. This is consistent
with parts of this region experiencing a small decrease in SST (Figure 3). Both SAI and MSB
yield changes in primary production comparable to that in RCP8.5 (19% and 18%
respectively), but where changes in radiation contribute ~2% to the total reduction. There is,
however, no corresponding change in cloud cover (see Muri et al., in prep.) to explain the
apparent importance of radiation changes in this region. The Indian Ocean is also one of the
regions where CCT able to sustain (i.e., induce least changes) the contemporary primary
production. After termination, the ocean primary production continues to decrease and is in
2121-2150 30% lower than in 1971-2000 in all experiments. Unusually, the temperature
changes lead to an increase in ocean primary production of 4% in 2121-2150 in all
experiments.



**3.6 Comparison with previous studies**
Very few other studies have been published on the impact on ocean biogeochemistry
due to RM, but two recent ones are Tjiputra et al. (2016) and Partanen et al. (2016). Tjiputra
et al. (2016) identified changes in ocean primary production and export production in a
simulation with SAI. The implementation of SAI is different here both in methodology and
amplitude of forcing, but the spatial signal of surface climate response and the overall impact
on global ocean primary production in broadly comparable. Nevertheless, our study provides
a more extended analysis in identifying the dominant drivers of changes in primary
production in key ocean regions. Partanen et al. (2016), on the other hand, analyzed the
effects on ocean primary production from MSB only. Overall, the effects of MSB in this
study and that of Partanen et al. (2016) are quite different both spatially and as a function of
time. This is likely due to the several noteworthy differences between their method and the
one used here:
(i) Partanen et al. (2016) uses the UVic ESCM model, an Earth system model of
intermediate complexity (EMIC) while here we use the fully coupled NorESM1-ME
Earth system model;
(ii) the RM forcing applied by Partanen et al. (2016) is -1 Wm$^{-2}$ annually, while here it is
scaled up to -4 Wm$^{-2}$ in 2100;
(iii) Partanen et al. (2016) applies RM to RCP4.5 while here we apply RM to RCP8.5;
(iv) Partanen et al. (2016) applies RM for 20 years before termination while here we
apply RM for 80 year before termination, which, combined with the higher forcing,
means that the Earth system takes longer to recover in this study than in the Partanen
et al. (2016) study.
The biggest and most important of these differences is that Partanen et al. (2016) use
an EMIC while we use an ESM. The ecosystem module in NorESM1-ME is not substantially



more complex than that of the UViC ESCM model, but differences could arise due to better
representation of the ocean physical circulation (owing to higher spatial resolution) and air-
sea interactions. Differences in the aerosol-cloud-climate interactions will also affect the
results. NorESM1-ME has a fully interactive tropospheric aerosol scheme, which is of key
importance when evaluating the impact of changes in shortwave radiation reaching the
surface from changes to clouds. Partanen et al. (2016) identify a decrease in global mean
ocean primary production relative to their reference case (RCP4.5) while in our MSB
simulation we simulate an increase in ocean primary production relative to our reference case
(RCP8.5). These differences and the large differences in the spatial impact can partly be
explained by the differences in the applied RM forcing and method, but is mostly explained
by the fundamental differences between the models and especially how clouds are modelled.
Another important difference between Partanen et al. (2016) and this study is the timing of
termination, since this is a very important aspect of all climate engineering studies. Partanen
et al. (2016) applies RM for 20 years before termination, while we apply RM for 80 years
before termination. This means that in our study the impact on temperature and ocean
circulation is greater than in the Partanen et al. (2016) study as the slow climate feedbacks are
allowed to pan out. This could explain the differences in termination effect between the
studies, where the primary production fully recovers and exceeds that in RCP4.5 in the
Partanen et al. (2016) study, but remain within the variability of RCP8.5 here. The larger
magnitude of the forcing applied in our simulations (-4 Wm$^{-2}$ in 2100) also means that it takes
much longer for the climate system to recover back to the RCP8.5 state.

**4 CONCLUSIONS**

In this study, we use the Norwegian Earth System Model with fully interactive carbon

cycle to assess the impact of three radiation management climate engineering (RM) methods




on marine biogeochemistry. The model simulations indicate that RM may reduce
perturbations in SST and thermocline oxygen driven by anthropogenic climate change, but
that large changes in primary production remain and are even intensified in some regions. It
must be noted that we use only one model, and that such models are known to have large
spread in their projections of future ocean primary production (*e.g.* Bopp et al., 2013).
However, this single-model study does show some clear tendencies:
(i)    A clear mitigation of the global mean decrease in ocean primary production from

10% in 2100 in RCP8.5 and ~5% in RCP4.5 to somewhere between 3% and 6%

depending on the method of RM.

(ii)   Strong regional variations in the changes, and what primarily drives the changes, in

ocean primary production. The different methods of RM do not have the same effects

in the same regions, even though SAI and MSB yield similar global averages.

(iii)  MSB yields the largest changes relative to RCP4.5, which is consistent with MSB

being applied over the ocean and therefore likely affects the ocean more strongly than

the other methods.

The effect of future climate change on ocean primary production is uncertain, and is
driven by an integrated change in physical factors such as temperature, radiation, and ocean
mixing. Additionally, changes in ocean oxygen concentrations and ocean acidification are
likely to affect ocean primary production. So it is noteworthy that with RM, anthropogenic
$CO_2$ emissions are not curbed, so ocean acidification would continue. The results presented in
this study show that future changes to ocean primary production would likely be negative on
average, but exhibit great variation both temporally and spatially, regardless of whether or not
RM is applied.




This study also show that for the first five to ten years after a sudden termination of
large-scale RM the SST, oxygen, surface pH, and primary production all experience changes
that are significantly larger than those projected without RM implementation or mitigation.
While there is still large uncertainty in how marine habitats respond to such rapid changes, it
is certain than they will have less time to adapt or migrate to a more suitable location and
potentially have higher likelihood to face extinction.
The results of this work does nothing to diminish the complexity of climate impacts on
primary production, but rather highlights that any change in ocean primary production is
driven by a combination of several variables which all change in different ways in the future,
and subsequently are affected differently when RM is applied. The importance of ocean
primary production for human societies, however, lies in its impact on food security in
general and fisheries in particular, for which regional changes are much more important than
global changes (Mora et al., 2013).

**ACKNOWLEDGEMENTS**
The authors acknowledge funding from the Norwegian Research Council through the project
EXPECT (229760). We also acknowledge NOTUR resource NN9182K, Norstore NS9033K
and NS1002K. Helene Muri was also supported by RCN project 261862/E10, 1.5C-BECCSy.
JT also acknowledges RCN project ORGANIC (239965). The authors want to thank Alf Grini
for his technical assistance in setting up and running model experiments and as well as the
rest of the EXPECT team





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

**FIGURES AND TABLES**
Figure 1. Time series of global average change in (a) oxygen content at 200-600m depth (%) and (b) SST (°C).
The oxygen change is relative to the 1971-2000 average in the historical run.



**Figure 2. The absolute change in oxygen concentration (200-600m) in 2071-2100 relative to 1971-2000 (in**
**moles $O_2$ m$^{-2}$). Panel (a) shows zonally averaged (in 2° latitude bands) change for all simulations. Global maps**
**of (b) RCP8.5, (c) RCP4.5, (d) RCP8.5 with SAI, (e) RCP8.5 with MSB, (f) RCP8.5 with CCT. Gray shading in b)-f)**
**indicates areas where the change is not significantly different from the 1971-2000 average (*i.e.* within one**
**standard deviation).**
**Figure 3. The absolute change in sea surface temperature (SST) in 2071-2100 relative to 1971-2000 (in °C).**
**Panel (a) shows zonally averaged (in 2° latitude bands) change for all simulations. Global maps of (b) RCP8.5,**
**(c) RCP4.5, (d) RCP8.5 with SAI, (e) RCP8.5 with MSB, (f) RCP8.5 with CCT. Gray shading in b)-f) indicates**
**areas where the change is not significantly different from the 1971-2000 average (*i.e.* within one standard**
**deviation).**
**Figure 4. Time series of global average change in (a) atmospheric $CO_2$ (ppm), (b) air-sea $CO_2$ flux (PgC yr$^{-1}$), (c)**
**global surface ocean pH, (d) global deep ocean (>2000 m) pH, and (e) deep (>2000 m) North Atlantic Ocean**
**(north of 30°N) pH.**
**Figure 5. Time series of changes global ocean primary production (PP, %). The primary production change is**
**relative to the 1971-2000 average in the historical run.**
**Figure 6. The percent changes in primary production in 2071-2100 relative to the 1971-2000 average in the**
**historical run. (a) zonally averaged (in 2° latitude bands) change for all simulations. (b) RCP8.5, (c) RCP4.5, (d)**
**RCP8.5 with SAI, (e) RCP8.5 with MSB, (f) RCP8.5 with CCT. Gray shading in b)-f) indicates areas where the**
**change is not significantly different from the 1971-2000 average (*i.e.* within one standard deviation). The**
**outlined areas in panel (b) indicate regions plotted in Figure 8.**
**Figure 7. Time series of the 5-year running mean of globally averaged primary production (PP, %) calculated**
**offline using Equation 1, plotted as the percent change relative to the 1971-2000 average in the historical**
**run. Note the different scales on the y-axes. See Table 1 for an explanation of the different calculations**
**shown.**
**Figure 8. Offline calculated primary production change (PP, %) in five different regions (as indicated on Figure**
**6b) for RCP4.5, RCP8.5, and RCP8.5 with three different RM methods.**
**Table 1. Description of the offline calculations of ocean primary production and its primary drivers using**
**Equation 1. T is temperature, L is shortwave radiation at the surface, N is the concentration of the limiting**
**nutrient (either nitrate, phosphate, silicate, or dissolved iron), and P is the concentration of phytoplankton**
**cells. $\overline{X}$ denotes the long-term (80 year) mean of the given variable.**

| Calculation | |
|---|---|
| Everything changes | T, L, N, P |
| Only temperature changes | T, $\overline{L}$, $\overline{N}$, $\overline{P}$ |
| Only shortwave radiation changes | L, $\overline{T}$, $\overline{N}$, $\overline{P}$ |
| Only phytoplankton concentration changes | P, $\overline{L}$, $\overline{N}$, $\overline{T}$ |

**Table 2. General description of model experiments used in this study.**

| Experiment | Description | Time period |
|---|---|---|
| RCP4.5 | Reference RCP4.5 scenario | 2006-2100 |
| RCP8.5 | Reference RCP8.5 scenario | 2006-2150 |
| SAI | RCP8.5 scenario where sulfur particles are injected into the atmosphere to scatter incoming shortwave radiation and bring down global average temperatures | 2020-2100 |
| SAI$_{EXT}$ | The extension of the SAI run after termination of climate engineering in 2100 | 2101-2150 |
| MSB | RCP8.5 scenario where salt particles are added to the marine boundary layer between 45°S and 45°N to make both the sky and clouds brighter, thus increasing | 2020-2100 |





| | the Earth's albedo thereby lower global average temperatures | |
|---|---|---|
| MSB$_{EXT}$ | The extension of the MSB run after termination of climate engineering in 2100 | 2101-2150 |
| CCT | RCP8.5 scenario where cirrus clouds are thinned out. Cirrus clouds have a net heating effect so thinner clouds will result in lower global average temperatures | 2020-2100 |
| CCT$_{EXT}$ | The extension of the CCT run after termination of climate engineering in 2100 | 2101-2150 |






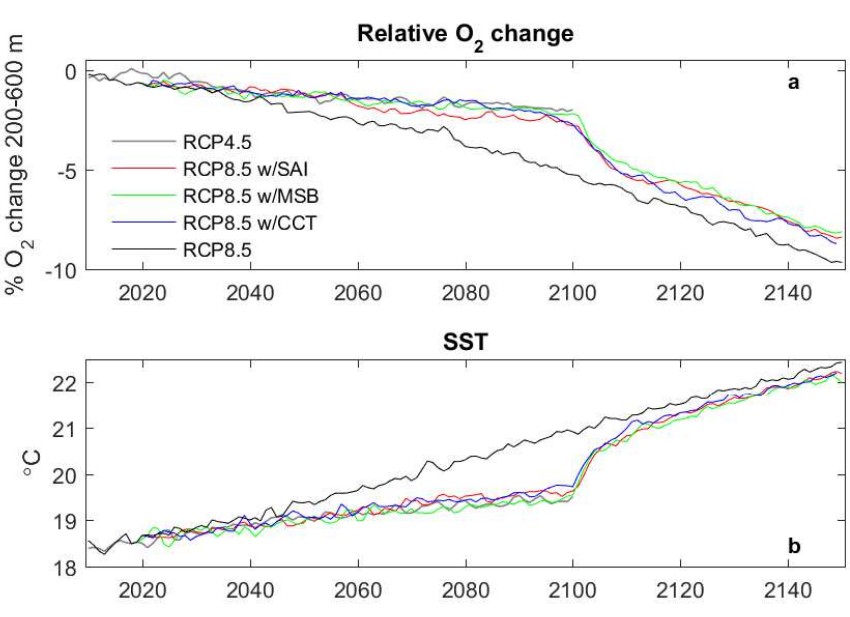

**Figure 1. Time series of global average change in (a) oxygen content at 200-600m depth (%) and (b) SST (°C). The oxygen change is relative to the 1971-2000 average in the historical run.**





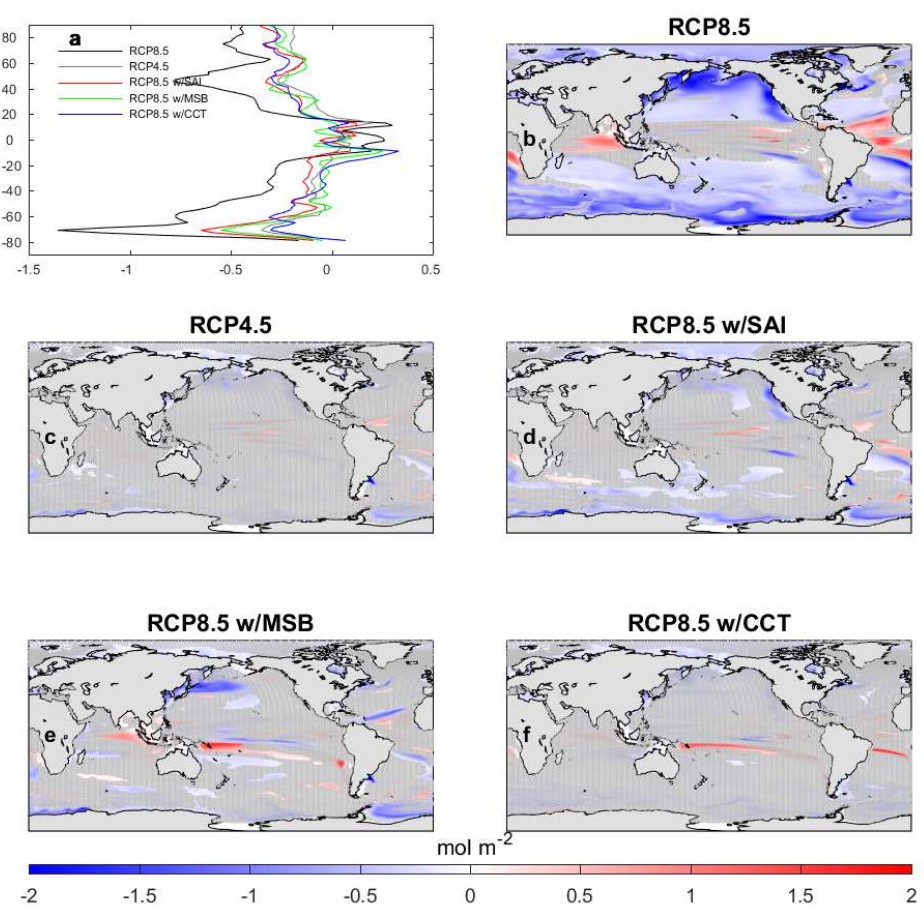

**Figure 2. The absolute change in oxygen concentration (200-600m) in 2071-2100 relative to 1971-2000 (in moles $O_2$ m$^{-2}$). Panel (a) shows zonally averaged (in 2° latitude bands) change for all simulations. Global maps of (b) RCP8.5, (c) RCP4.5, (d) RCP8.5 with SAI, (e) RCP8.5 with MSB, (f) RCP8.5 with CCT. Gray shading in b)-f) indicates areas where the change is not significantly different from the 1971-2000 average (*i.e.* within one standard deviation).**




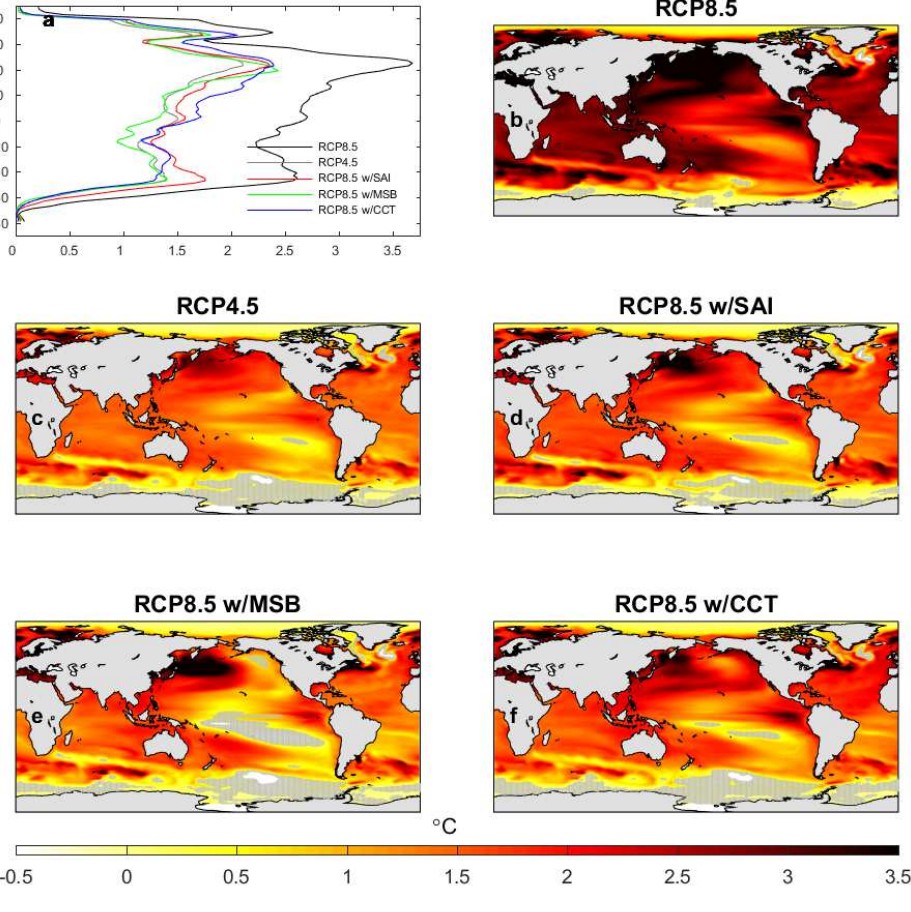

**Figure 3.** The absolute change in sea surface temperature (SST) in 2071-2100 relative to 1971-2000 (in °C). Panel (a) shows zonally averaged (in 2° latitude bands) change for all simulations. Global maps of (b) RCP8.5, (c) RCP4.5, (d) RCP8.5 with SAI, (e) RCP8.5 with MSB, (f) RCP8.5 with CCT. Gray shading in b)-f) indicates areas where the change is not significantly different from the 1971-2000 average (*i.e.* within one standard deviation).



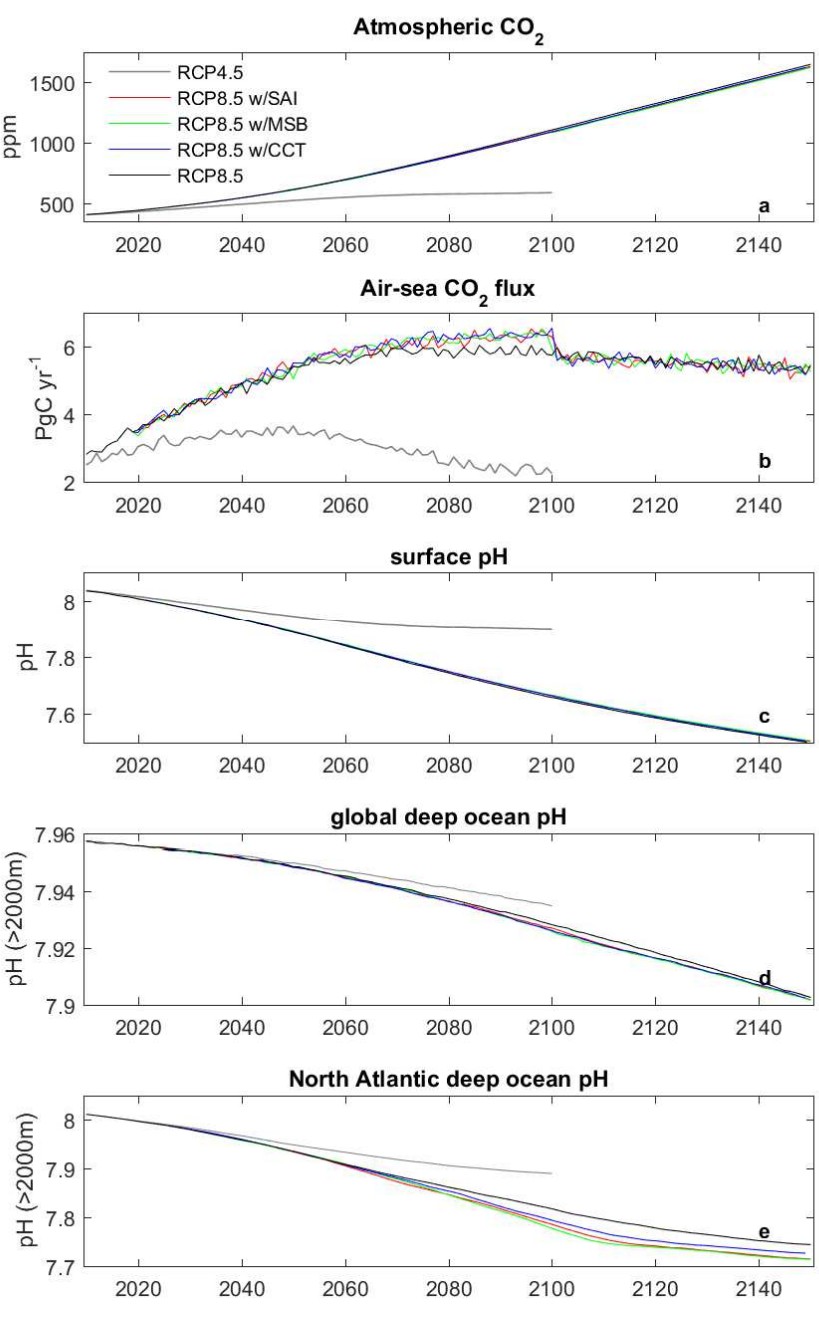

**Figure 4.** Time series of global average change in (a) atmospheric $CO_2$ (ppm), (b) air-sea $CO_2$ flux (PgC yr$^{-1}$), (c) global surface ocean pH, (d) global deep ocean (>2000 m) pH, and (e) deep (>2000 m) North Atlantic Ocean (north of 30°N) pH.





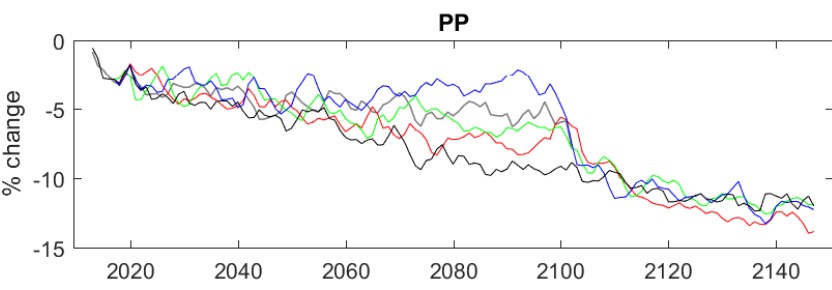

**Figure 5. Time series of changes global ocean primary production (PP, %). The primary production change is relative to the 1971-2000 average in the historical run.**





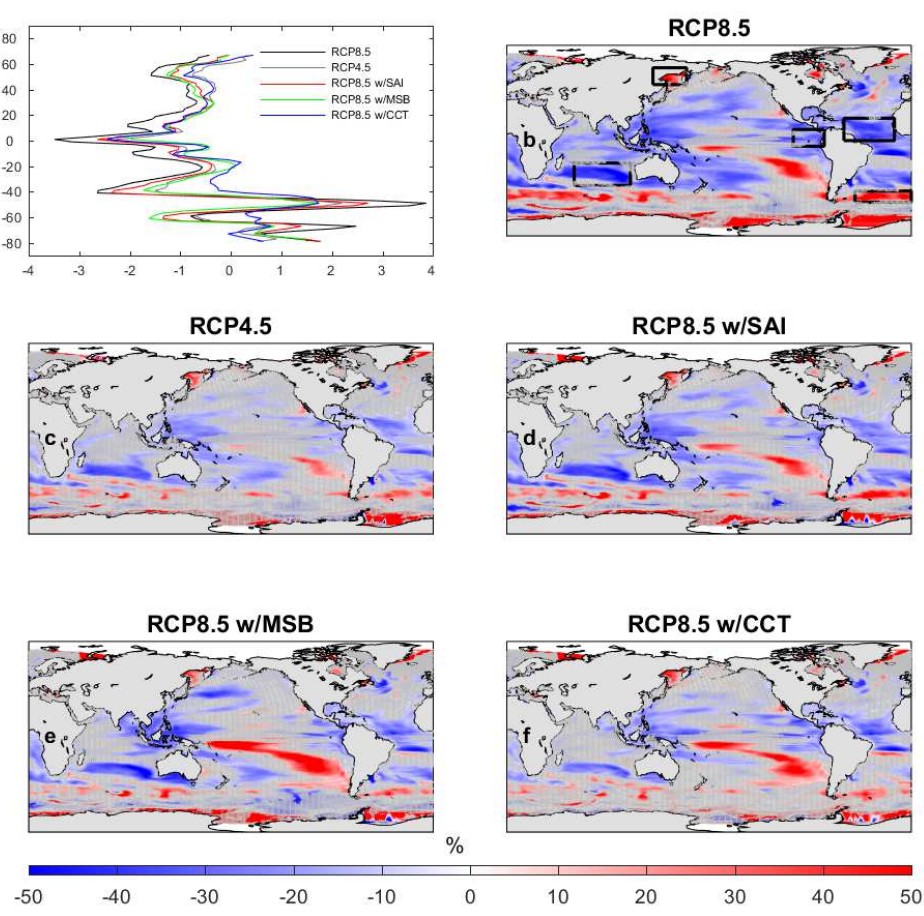

**Figure 6.** The percent changes in primary production in 2071-2100 relative to the 1971-2000 average in the historical run. (a) zonally averaged (in 2° latitude bands) change for all simulations. (b) RCP8.5, (c) RCP4.5, (d) RCP8.5 with SAI, (e) RCP8.5 with MSB, (f) RCP8.5 with CCT. Gray shading in b)-f) indicates areas where the change is not significantly different from the 1971-2000 average (*i.e.* within one standard deviation). The outlined areas in panel (b) indicate regions plotted in Figure 8.



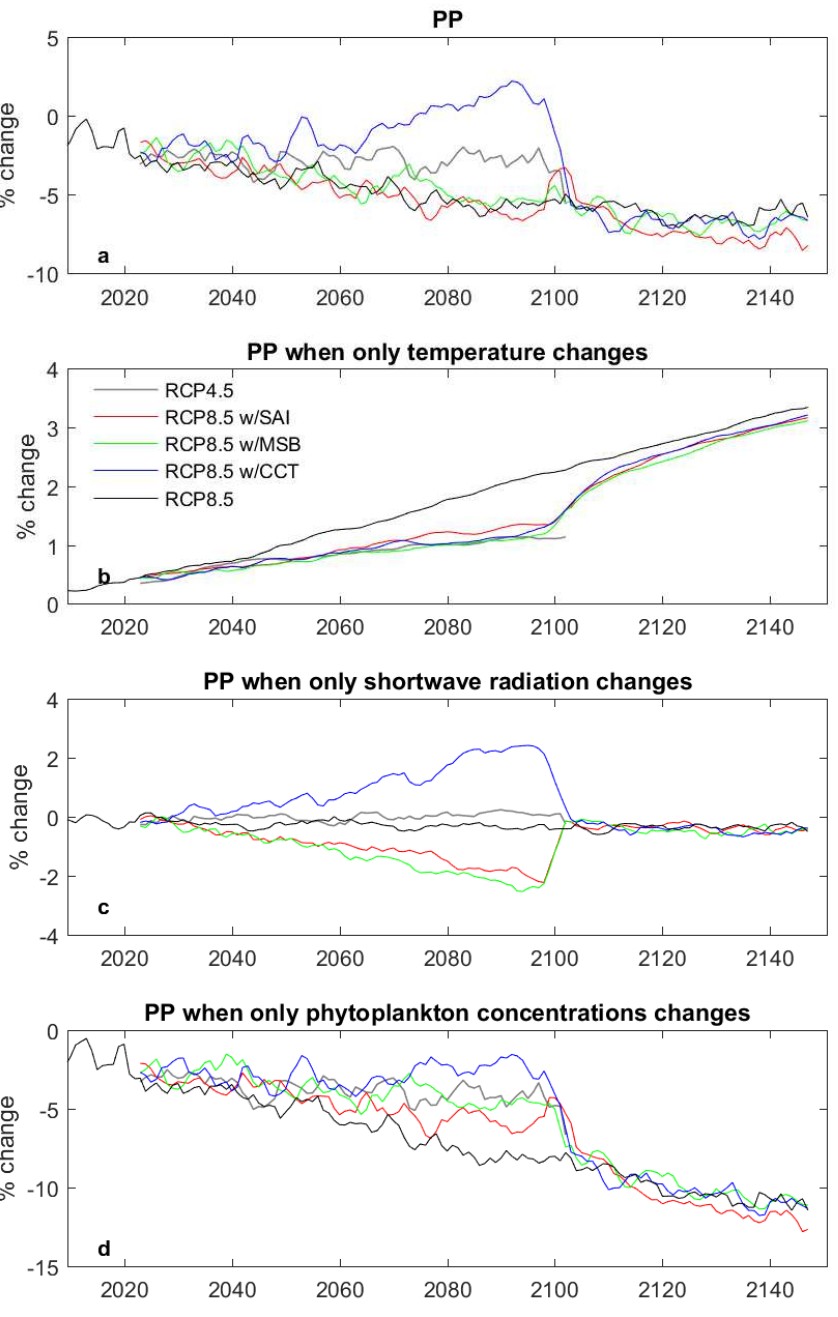

**Figure 7.** Time series of the 5-year running mean of globally averaged primary production (PP, %) calculated offline using Equation 1, plotted as the percent change relative to the 1971-2000 average in the historical run. Note the different scales on the y-axes. See Table 1 for an explanation of the different calculations shown.




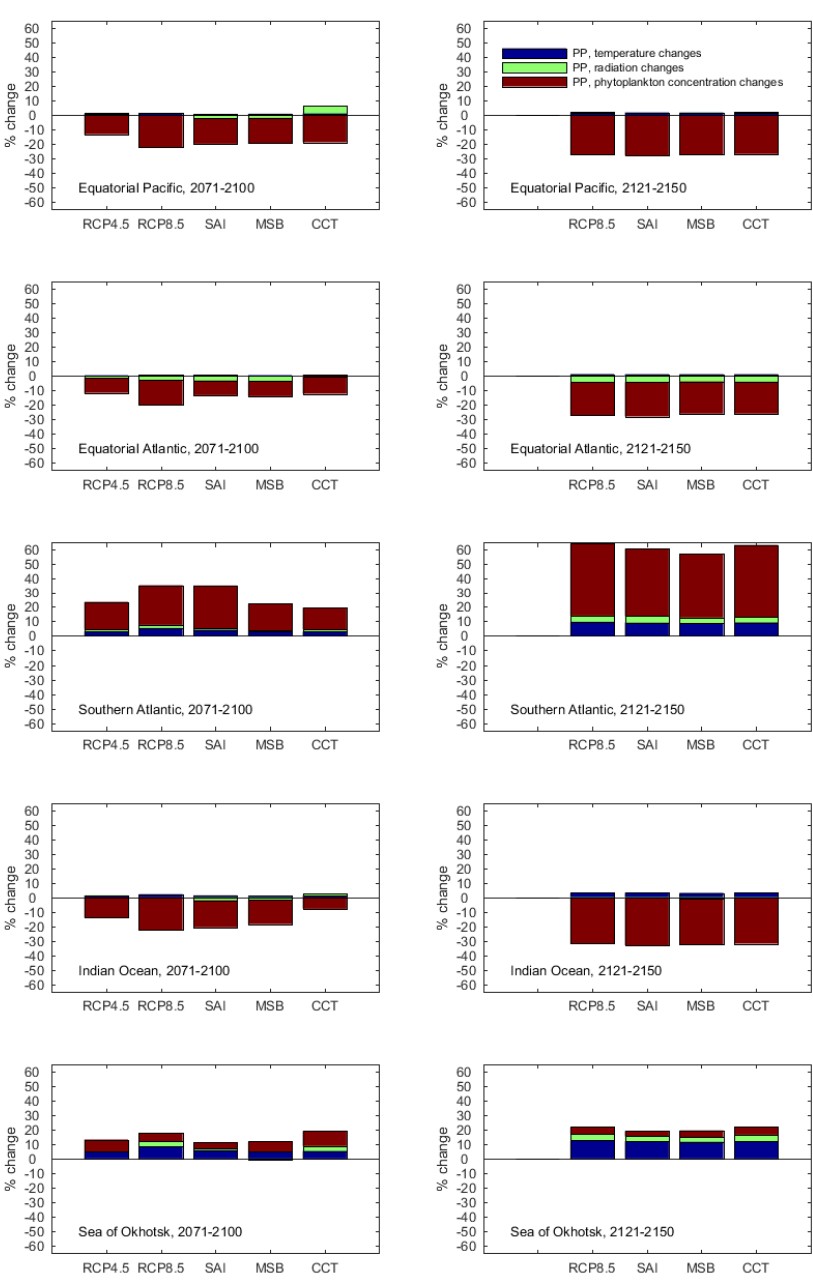

**Figure 8.** Offline calculated primary production change (PP, %) in five different regions (as indicated on Figure 6b) for RCP4.5, RCP8.5, and RCP8.5 with three different RM methods.