# Peer review of "Climate engineering and the ocean: effects on biogeochemistry and primary production"

_Biogeosciences, 2017_

## Short Comment (SC1) · 16 Jun 2017

In reaching their abstract's conclusion, ave the authors considered and compared to SRM models circumglobal natural albedo variations like those seen in the calcite belt or created by the extensive summer plactonic blooms that annually cover a significant percentage of Northern seas? The radiative forcings resulting from them may rival or exceed humanity's continental albedo footprint

[Figure]

[Figure]

**Fig. 1.**

---

## Referee Comment (RC1) · Anonymous Referee #1 · 12 Jul 2017

General comments

"Climate engineering and the ocean: effects on biogeochemistry and primary production" by Lauvset et al. provides a single-model assessment how three different climate engineering methods (stratospheric aerosol injections, marine sky brightening and cirrus cloud thinning) affect ocean biogeochemistry. This is one of the first studies on the topic and comparing different methods within the same model is a valuable addition to previous works. They concentrate on four key variables in ocean biogeochemistry: sea surface temperature, oxygen, pH and net primary production. For NPP, they complement the interactive Earth System Model simulations with offline calculations that make possible to disentangle different drivers of NPP change. This method adds to the value of the manuscript, although I have some concerns and questions about the

method (see specific comments).

The manuscript is mostly clearly structured and written, and thus easy to read. However, some more commas when dependent clauses start sentences would enhance readability. For example, I would insert a comma in "When only phytoplankton concentration is allowed to vary temporally in the offline calculation there is a decrease of ∼8% by 2100 in RCP8.5." (Lines 369-371) and similar sentences. Also, the use of present tense throughout the manuscript differs from the general practice of using past tense to describe the results and methods. Overall, I would recommend this manuscript for publication if my comments below are adequately addressed.

Specific comments

Major comments

The offline model for NPP calculations needs more precise explanation and evaluation. In Lines 139-149, you imply that monthly-mean values are used for nutrients. On the other hand, on Lines 362-364 you write that phytoplankton concentration is used as a proxy for nutrient availability. Moreover, on Line 417, phytoplankton concentration is said to be a proxy for circulation changes. The last two statements are in my understanding consistent with each other (but it would be good to explain explicitly why they are related), but please clarify how the first statement of monthly-mean nutrient fields should be understood.

Also, doesn't NPP significantly affect phytoplankton concentration? Using phytoplankton concentration to calculate NPP sounds circular reasoning to me and I see a risk that the method overestimates the contribution of circulation changes to NPP changes. For example, if temperature increased phytoplankton in the online simulations and this in turn increases NPP in offline calculations, don't you attribute this increase to circulation in the offline calculations instead of to temperature?

I think it would also be good to provide some short evaluation of the offline NPP calculation method to show whether it provides similar results as the online calculation. The value of offline calculations is to disentangle different drivers of NPP change, but how well does the offline version compare to online version when all drivers are accounted for (both regionally and at global mean level)? Specifically, comparing Fig. 5 to Fig. 7a would be helpful.

Minor comments

Lines 20-22: If the drivers of NPP are "partly" affecting the inhomogeneity of the NPP changes, what is responsible for the rest of the inhomogeneity?

Line 93: Spell out SST as it's used here for the first time.

Line 118 and throughout the manuscript: You apparently use NPP and primary production interchangeably. I would recommend using NPP (shorter and more precise) everywhere consistently or explain if there is some subtle difference between NPP and primary production in the manuscript.

Line 165: I think it would more precise to say that you scaled AOD to match the level of a 20 TgS/year injection as you don't explicitly model the aerosol injection here.

Line 172: Maybe good to say here explicitly that the other two methods had -4.0 W m-2 forcing.

Line 193: SST should be defined on Line 93 already. Maybe not necessary to repeat it here.

Lines 207-209: You use a high emission scenario. I would add that RM does not prevent long-term impacts in a scenario where CO2 emissions don't go to net zero. If they did, the situation would probably look a lot different.

Lines 230-232: Are there many areas where changes are greater with RM than without? If the results in RCP8.5 with RM are spatially highly variable, the changes can't be attributed to RM.

Lines 291-292: I'm not sure what this sentence means. What is smaller than in RCP8.5? The exhibited decrease of NPP or the changes in NPP in RM simulations? Please, clarify.

Line 332-334: Isn't the increase in NPP with CCT only present in offline calculations? In Fig. 5, NPP decreases in all simulations, and I think the online calculations are more reliable.

Line 363: As discussed earlier, please explain here or elsewhere what you mean by using phytoplankton as a proxy for nutrient availability.

Line 378: Is this section based on online of offline NPP calculations? If you use only offline calculations, could you provide some evaluation how well the offline results match the online results at regional level?

Line 388-390: What do you exactly mean by being consistent with CMIP5? Consistent with the sign of model ensemble mean or do all CMIP5 models give the same sign for these regions?

Lines 403-409. Why higher NPP would not lead to higher fish catches but lower NPP would decrease fish catches? Is this based on some dynamics of the ecosystem or are you just more careful to predict any increases than to predict decreases?

Lines 411-414: Splitting this to several sentences would make it easier to understand. Also, "do" on Line 413 seems redundant.

Line 422: I don't understand what you mean by "Radiation changes become more important in driving changes with RM".

Line 463: Why is this unusual? Compared to what? Doesn't increased temperature lead to increased NPP in other regions as well?

Line 467: Considering the low number of previous studies on the topic, could you write something about the results of Hardman-Mountford et al (2013) that you mention in the

introduction? I know that comparing an ESM to single-column model is challenging, but it would be interesting to know how the results compare.

Lines 494-497: I would add here that the potential interaction of SST and the clouds is missing in Partanen et al. (2016). Their forcing is calculated with an AGCM that has a fully interactive aerosol scheme and takes thus into account interactions with clouds and sea salt aerosol, but with prescribed SST, the model might miss some relevant feedbacks.

Lines 497-500: Could you speculate, what are the implications of using a high emission scenario (RCP8.5) instead of a low emission scenario (RCP4.5)? Table 2: I would write that AOD is modified to reflect a sulphur injection not to give an impression that the sulphur injection is calculate online in the current study.

Figure 2 and other maps: Could you move labels a,b,c,… outside the plots? They are a bit hard to see and I first thought they were missing altogether. All line plots: The lines are a bit hard to tell apart. I know that with so many overlapping lines it's hard to make them easy to distinguish, but I think there could be some room for improvement using dashed lines or slightly thicker lines or something.

Figure 5. The legend is missing. Also, why is there a gap in the line of CCT around 2100?

Figure 6: Standard deviation of what? Inter-annual variability of annual means of the reference period?

Figure 7. Could the legend be included in sub figure a already?

Technical corrections

Line 34: temperatures -> temperature

Line 39: I think "induced" is redundant here.

Line 235: continue -> continues (if you keep the present tense)

Line 408: decreases -> decrease

Lines 472-473: A verb is missing. (in -> are ?)

———————————————————

---

## Referee Comment (RC2) · Anonymous Referee #2 · 4 Sep 2017

The manuscript by Lauvset at al. analyses the effects of three proposed solar radiation schemes for geo-engineering on ocean carbon cycling (CC) and net primary productivity (NPP), using a fully coupled earth system model which includes an aerosol and a radiation scheme, a description of atmospheric and oceanic circulation, and land and ocean biogeochemical models. The question investigated is highly relevant, both for understanding possible feedbacks in the system (changes in radiative climate forcing incurred by changes in oceanic carbon uptake) and for possible effects of (engineered or un-engineered) climate change on food security: primary production of the ocean can serve as a (admittedly crude) measure of possible fisheries yields. Three geo-engineering schemes, all affecting the radiation balance, two mainly on the incoming shortwave radiation, and the third mainly on the outgoing long-wave radiation are ap-

plied in this study, in such a way that globally they all lead to a reduction of the radiative flux by 4 W m$^2$, bringing the radiative forcing of the RCP8.5-scenario down to that of RCP4.5. In addition to these coupled model runs, the manuscript uses offline calculations to investigate which factors drive changes in NPP. These help in interpreting the results, but as outlined further below I have some issues with the methodology here.

Overall, this is a well thought-through study, the results are relevant, and the manuscript is besides some minor points very well written. I would therefore support publication in Biogeosciences after addressing the points listed below.

**Major comments**

The description of the offline calculations (lines 139 ff) is missing important information, and also some justification. To me it is not clear at all to which equations the expression 'makes use of the same set of equations as the online calculation' (line 141) refer to: Does the offline model consider three-dimensional transport (advection and diffusion) of the non-prescribed equations? Which equations exactly are those? Why is the light in the offline calculations attenuated to a constant depth of 50 m, is the offline model two-dimensional or does it resolve depth?

One issue that I found particularly confusing in the description of the offline experiments is that N stands for the most-limiting nutrient (phosphate/nitrate/iron). But which nutrient is most limiting is likely to change in the online runs. Are all nutrients prescribed in the offline runs, is there a climatology of the most limiting nutrient?

I also have a similar problem with the interpretation of the results of the offline calculations as the first reviewer. The authors use phytoplankton biomass as proxy for assessing the impact of changes in nutrient supply to the euphotic zone due to changes in upper ocean stratification (lines 363-364). What one would really like to use as a control variable in these calculations is the vertical flux of nutrients. I see that nutrient concentrations are probably not a good tracer for this nutrient flux, since they are drawn down to limiting values (assuming sufficient light) regardless of the flux. But

the phytoplankton biomass is also just an indirect indicator: Firstly it is also affected by other losses such as zooplankton grazing (as the authors also mention, line 366), to which I would add the sinking losses of biomass through aggregation and sinking: Assume that the only loss of phytoplankton was a quadratic loss through aggregation and sinking. Then biomass would be proportional to the square root of nutrient supply. Also, phytoplankton growth rate is affected by both nutrients and temperature, which however is considered as a separate driver. To me it is thus nor completely clear how well these two factors can be separated with the offline experiments.

A smaller question that I didn't find the answer to in the model description (lines 129-138), and that may affect the interpretation of the manuscript slightly, is whether the model considers direct effects of ocean acidification (line 536) on carbon cycling through the marine ecosystem, e.g. by reductions in calcification.

Also, the description of how the different RM methods have been implemented in the model (Lines 163-173) is quite short: to me it was for example a bit unclear how the SAI scenario was modelled. It is said that a layer of sulfate aerosols was prescribed, but then the next sentence states an injection strength, which to me implies that the layer was not prescribed, but calculated as resulting from a balance between injection and some unclear losses.

**Minor comments**

Line 42: At least the CCT method does not act to 'increase the amount of solar radiation reflected' but rather to increase the loss of long-wave radiation passing through the atmosphere.

Line 66 ff: I found this sentence quite confusing: Is it maybe two sentences in one?

Line 100: contrary to the statement on line 100 I have not found any presentation of impacts on inorganic carbon in the manuscript, only impacts on air-sea carbon flux. They are of course closely related, but be precise.
Line 138: It is stated that seawater carbonate chemistry formulation follows the OCMIP protocol. But which one, OCMIP 2 or 3? OCMIP 3 corrected a few smaller errors in the OCMIP 2 protocols.

Line 223-225: This result could be emphasised a bit more, it shows why we need full coupled atmosphere-ocean-biogeochemistry models to study this type of effects

Line 297: 'production' missing after 'increasing primary'

Line 299-300: 'after termination it takes less than 5 years': What sets the timescale, the atmosphere (radiation), or the ocean biology?

Line 327: 'Only CCT significantly changes..': Does that not contradict what has been said before? Maybe I did not understand what should be said here.

Line 336-337: insert 'the' in 'once terminated, CCT method..'

Line 441: Is 18 percent really a 'minor change' compared to 13 percent?

Line 447 ff: This and the next paragraph talk about reduction on NPP; it would be clearer if the percent changes would therefore have a negative sign also.

Line 477: 'are quite different': It would be good to have a short summary of the differences, so the reader does not have to read Partanen et al. (2016) herself.

Line 563 ff, references: It the Ahlm paper still in the discussion forum or is there a citable full reference by now?

---

## Author Comment (AC1) · 29 Sep 2017

Thank you for this interesting comment on an additional aspect this type of research. In our study we have only evaluated the effects and impacts of artificial albedo changes in the form of radiation management in an Earth System model. The model we use, NorESM1-ME, also does not include albedo changes due to changes in plankton blooms so this is not possible to study with this particular model.

---

## Author Comment (AC2) · 29 Sep 2017

General comments
"Climate engineering and the ocean: effects on biogeochemistry and primary production" by Lauvset et al. provides a single-model assessment how three different climate engineering methods (stratospheric aerosol injections, marine sky brightening and cirrus cloud thinning) affect ocean biogeochemistry. This is one of the first studies on the topic and comparing different methods within the same model is a valuable addition to previous works. They concentrate on four key variables in ocean biogeochemistry: sea surface temperature, oxygen, pH and net primary production. For NPP, they complement the interactive Earth System Model simulations with offline calculations that make possible to disentangle different drivers of NPP change. This method adds

to the value of the manuscript, although I have some concerns and questions about the method (see specific comments). The manuscript is mostly clearly structured and written, and thus easy to read. However, some more commas when dependent clauses start sentences would enhance readability. For example, I would insert a comma in "When only phytoplankton concentration is allowed to vary temporally in the offline calculation there is a decrease of $\tilde{8}\%$ by 2100 in RCP8.5." (Lines 369-371) and similar sentences. Also, the use of present tense throughout the manuscript differs from the general practice of using past tense to describe the results and methods. Overall, I would recommend this manuscript for publication if my comments below are adequately addressed.

**Thank you for this nice summary and comments about the manuscript. Since the results and discussion are combined into one section we feel that present tense is the most appropriate. The tense has been changed in the methods section.**

Major comments
The offline model for NPP calculations needs more precise explanation and evaluation. In Lines 139-149, you imply that monthly-mean values are used for nutrients. On the other hand, on Lines 362-364 you write that phytoplankton concentration is used as a proxy for nutrient availability. Moreover, on Line 417, phytoplankton concentration is said to be a proxy for circulation changes. The last two statements are in my understanding consistent with each other (but it would be good to explain explicitly why they are related), but please clarify how the first statement of monthly-mean nutrient fields should be understood.

**Upon rereading these sections we see that our description of both the method and the interpretation of results could have been better. We believe some of the confusion comes from the difference between phytoplankton growth rate and primary production, and the text has been revised to clarify this. The growth rate of phytoplankton is a function of temperature, light, and the concentration**

**of the limiting nutrient (in our case either nitrate, phosphate, or dissolved iron). The growth rate is expressed as the first two terms in Equation 1 in the original paper [r(T,L)*(N/(N+N0)]. In this equation, monthly mean nutrient data from the model are used. As is seen from this formulation, any change in the limiting nutrient has a very small impact on the growth rate. NPP is the growth rate multiplied by the phytoplankton biomass (expressed as a concentration), i.e. Equation 1 in entirety. To help clarify this we have, in the revised paper, split Equation 1 into one equation for growth rate and one for NPP.**

Also, doesn't NPP significantly affect phytoplankton concentration? Using phytoplankton concentration to calculate NPP sounds circular reasoning to me and I see a risk that the method overestimates the contribution of circulation changes to NPP changes. For example, if temperature increased phytoplankton in the online simulations and this in turn increases NPP in offline calculations, don't you attribute this increase to circulation in the offline calculations instead of to temperature?

**The reviewer is correct, and we appreciate this being pointed out. As described above, NPP is driven by temperature, light, nutrient and phytoplankton concentrations. Since the last two drivers depend on each other, in the revised manuscript, we have quantified the changes in NPP (i.e., through the offline calculation) due to changes in temperature, light, and residual parameters. The residual term is approximately represents an integrated circulation-induced changes in phytoplankton and limiting nutrient as described in the revised manuscript. We believe this will avoid confusions on the 'circular effects' as the reviewer pointed out.**

I think it would also be good to provide some short evaluation of the offline NPP calculation method to show whether it provides similar results as the online calculation. The value of offline calculations is to disentangle different drivers of NPP change, but how well does the offline version compare to online version when all drivers are

accounted for (both regionally and at global mean level)? Specifically, comparing Fig. 5 to Fig. 7a would be helpful.

**We agree with the reviewer that comparing the offline NPP with the online is useful. Given the method used to calculate NPP offline (see my reply above) we expect there to be some differences between the offline and online estimates. Fig 1 below shows a comparison between the 2006-2020 NPP in the model and the 2006-2020 NPP calculated offline. In 2020, the offline global average NPP is 75In the five regions we discuss in more depth the percent change in 2071-2100 relative to 1971-2000 differs by 1-9% between online and offline NPP. A new figure, Figure 8, identical to Figure 6 but plotted using the offline NPP, has been added to the manuscript. Some text has also been added to clarify these differences and make clear throughout the discussion where NPP is being discussed.**

Minor comments

Lines 20-22: If the drivers of NPP are "partly" affecting the inhomogeneity of the NPP changes, what is responsible for the rest of the inhomogeneity?

**We agree with the reviewer that this sentence was unclear and have revised to: "The spatially inhomogeneous changes in ocean NPP are related to the simulated spatial change in the NPP drivers (incoming radiation, temperature, availability of nutrients, and phytoplankton) depending in the RM methods." In addition, we have added some text with concrete examples of how the different RM methods affect NPP differently.**

Line 93: Spell out SST as it's used here for the first time. **Done**

Line 118 and throughout the manuscript: You apparently use NPP and primary production interchangeably. I would recommend using NPP (shorter and more precise)

everywhere consistently or explain if there is some subtle difference between NPP and primary production in the manuscript. **Done**

Line 165: I think it would more precise to say that you scaled AOD to match the level of a 20 TgS/year injection as you don't explicitly model the aerosol injection here. **The text has now been clarified to: "As the NorESM1-M model does not include an interactive aerosol scheme in the stratosphere, the dataset of Tilmes et al. (2015) was used. The stratospheric zonal aerosol extinction, single scattering albedo and asymmetry factors resulting from SO2 injections in the tropics were prescribed such that the prescribed aerosol layer in year 2100 corresponds to an SO2 injection strength of 40 Tg yr-1 (Muri et al. 2017)."**

Line 172: Maybe good to say here explicitly that the other two methods had -4.0 W m-2 forcing. **Done**

Line 193: SST should be defined on Line 93 already. Maybe not necessary to repeat it here. **Done**

Lines 207-209: You use a high emission scenario. I would add that RM does not prevent long-term impacts in a scenario where CO2 emissions don't go to net zero. If they did, the situation would probably look a lot different. **Done.**

Lines 230-232: Are there many areas where changes are greater with RM than without? If the results in RCP8.5 with RM are spatially highly variable, the changes can't be attributed to RM. **We are unsure what the reviewer asks here since Figures 2, 3, and 6 all show the spatial variability in changes incurred by adding RM to RCP8.5. As shown**

**in Figures 2 and 3, RM induced changes are always smaller, or in a few cases in the opposite direction, than the results in the RCP8.5 reference simulation. We have rephrased "(. . .) possibly lead to new and detrimental (. . .)" to now read "(. . .) still lead to similar albeit weaker detrimental (. . .)"**

Lines 291-292: I'm not sure what this sentence means. What is smaller than in RCP8.5? The exhibited decrease of NPP or the changes in NPP in RM simulations? Please, clarify.
**The temporal decrease in global ocean NPP is smaller in experiments with RM than in RCP8.5. The sentence has been rewritten for clarity and now reads: "All RM methods also exhibit decreases in ocean NPP, but the decrease is never as strong as that in RCP8.5."**

Line 332-334: Isn't the increase in NPP with CCT only present in offline calculations? In Fig. 5, NPP decreases in all simulations, and I think the online calculations are more reliable.
**Yes, this is present only in the offline calculations and it is right that the online calculations are more "correct". However, on lines X-Y (previously 332-334) it is the results from the offline calculation that are being discussed. This is now clarified in the text which now reads: "In fact, CCT results in an increased productivity by 2100 (Figure 7a) in the offline calculation". While we agree that this statement was misplaced, we maintain that the effect of CCT on NPP is an interesting result and have moved this discussion to section 3.3.**

Line 363: As discussed earlier, please explain here or elsewhere what you mean by using phytoplankton as a proxy for nutrient availability. **See earlier reply.**

[Figure]

Line 378: Is this section based on online of offline NPP calculations? If you use only offline calculations, could you provide some evaluation how well the offline results match the online results at regional level?

**Since the online NPP cannot be decomposed into its individual drives this section is based entirely on the offline calculations. This is clarified in the text, which now reads "For a more detailed analysis, five regions have been identified and analyzed based on the offline calculations of NPP and its drivers." We have evaluated how the offline calculated NPP compares to the online model output. Depending on region, the total percent change in 2071-2100 relative to 1971-2000 differs by 1-9% between online and offline. The online change is higher in 3 of the 5 regions, while offline changes are higher in the remaining 2 regions. The new Figure 8 allows for comparison between the spatial variations of the online and offline NPP.**

Line 388-390: What do you exactly mean by being consistent with CMIP5? Consistent with the sign of model ensemble mean or do all CMIP5 models give the same sign for these regions?

**Our results are consistent with the CMIP5 model ensemble mean. This has been clarified in the text.**

Lines 403-409. Why higher NPP would not lead to higher fish catches but lower NPP would decrease fish catches? Is this based on some dynamics of the ecosystem or are you just more careful to predict any increases than to predict decreases?

**NPP is the building block of the food web. It is therefore straight forward to predict that if this decreases there is less food for all higher trophic levels. It is not, however, as straight forward to predict what happens to higher tropic levels if NPP increases. In addition, higher tropic levels in the ocean is more than just fish. We have reworded this section for clarity, and added the following**

statement: "The IPCC-AR5 states that due to lack of consistent observations it remains uncertain how the future changes in marine ecosystem drivers (like productivity, acidification, and oxygen concentrations) will alter the higher trophic levels (Pörtner et al., 2014)."

Lines 411-414: Splitting this to several sentences would make it easier to understand. Also, "do" on Line 413 seems redundant. **Done**

Line 422: I don't understand what you mean by "Radiation changes become more important in driving changes with RM".
**The reviewer is correct that this was a poorly worded sentence. The sentence is now revised for clarity and reads: "When RM is applied, shortwave radiation changes at the surface become more important in driving NPP changes than they are in RCP8.5 and RCP4.5".**

Line 463: Why is this unusual? Compared to what? Doesn't increased temperature lead to increased NPP in other regions as well?
**The unusual part is how large the temperature component is. The sentence has been revised for clarity and now reads: "The temperature changes lead to an unusually large, compared to other regions, increase in ocean NPP of 4% in 2121-2150 in all experiments."**

Line 467: Considering the low number of previous studies on the topic, could you write something about the results of Hardman-Mountford et al (2013) that you mention in the introduction? I know that comparing an ESM to single-column model is challenging, but it would be interesting to know how the results compare.
**A brief description of the Hardman-Mountford et al (2013) results and how they compare with our study has been added at the beginning of section 3.6 (before the comparison with Partanen et al (2016)).**

Lines 494-497: I would add here that the potential interaction of SST and the clouds is missing in Partanen et al. (2016). Their forcing is calculated with an AGCM that has a fully interactive aerosol scheme and takes thus into account interactions with clouds and sea salt aerosol, but with prescribed SST, the model might miss some relevant feedbacks.
**Thank you for pointing this out. A comment on this has been added: "Partanen et al. (2016) take their SRM forcing from Partanen et al. (2012), which use an atmosphere only version of their model and hence neglect important feedbacks, including SST/ocean feedbacks. Partanen et al. (2016) furthermore prescribe their SRM forcing in terms of changes to the radiation, and hence miss out on further feedbacks, that we include in our fully coupled Earth system simulations. E.g., as seen in Ahlm et al., (2017) and Muri et al. (2017), MSB may lead to an increased sinking of air over the oceans and hence a reduction in cloud cover."**

Lines 497-500: Could you speculate, what are the implications of using a high emission scenario (RCP8.5) instead of a low emission scenario (RCP4.5)?
**Generally, the global mean and rate of change of ecosystem drivers in RCP4.5 are smaller than RCP8.5 (Henson et al., 2017). Applying the same RM forcing on RCP4.5 projection would yield a global mean state that is closer to the pre-industrial state with model-dependent regional variations. A short sentence has been added reflecting this.**

Table 2: I would write that AOD is modified to reflect a sulphur injection not to give an impression that the sulphur injection is calculate online in the current study. **The table has been updated with a more precise definition of the experiments.**

Figure 2 and other maps: Could you move labels a,b,c,... outside the plots? They are a bit hard to see and I first thought they were missing altogether. **Done**

All line plots: The lines are a bit hard to tell apart. I know that with so many overlapping lines it's hard to make them easy to distinguish, but I think there could be some room for improvement using dashed lines or slightly thicker lines or something. **We have altered the figures slightly so that they now, hopefully, are easier to read.**

Figure 5. The legend is missing. Also, why is there a gap in the line of CCT around 2100?
**The gap is a glitch in the making of a png figure, it does not exist in the higher quality pdf figure. The pdf version is attached to this reply (Fig 2 below) and will be included in the revised submission. The legend is added.**

Figure 6: Standard deviation of what? Inter-annual variability of annual means of the reference period?
**One standard deviation is defined as the standard deviation of the mean of the 1971-2000 period in the historical run. This is now clarified in the text and in all relevant figure captions.**

Figure 7. Could the legend be included in sub figure a already? **Done.**

Technical corrections – **All have been changed accordingly.**
Line 34: temperatures -> temperature Line 39: I think "induced" is redundant here. Line 235: continue -> continues (if you keep the present tense) Line 408: decreases -> decrease Lines 472-473: A verb is missing. (in -> are ?)

[Figure]

**Fig. 1.**

[Figure]

**Fig. 2.**

---

## Author Comment (AC3) · 29 Sep 2017

The manuscript by Lauvset at al. analyses the effects of three proposed solar radiation schemes for geo-engineering on ocean carbon cycling (CC) and net primary productivity (NPP), using a fully coupled earth system model which includes an aerosol and a radiation scheme, a description of atmospheric and oceanic circulation, and land and ocean biogeochemical models. The question investigated is highly relevant, both for understanding possible feedbacks in the system (changes in radiative climate forcing incurred by changes in oceanic carbon uptake) and for possible effects of (engineered or un-engineered) climate change on food security: primary production of the ocean can serve as a (admittedly crude) measure of possible fisheries yields. Three geoengineering schemes, all affecting the radiation balance, two mainly on the

incoming shortwave radiation, and the third mainly on the outgoing long-wave radiation are applied in this study, in such a way that globally they all lead to a reduction of the radiative flux by 4 W m2, bringing the radiative forcing of the RCP8.5-scenario down to that of RCP4.5. In addition to these coupled model runs, the manuscript uses offline calculations to investigate which factors drive changes in NPP. These help in interpreting the results, but as outlined further below I have some issues with the methodology here. Overall, this is a well thought-through study, the results are relevant, and the manuscript is besides some minor points very well written. I would therefore support publication in Biogeosciences after addressing the points listed below.

Major comments
The description of the offline calculations (lines 139 ff) is missing important information, and also some justification. To me it is not clear at all to which equations the expression 'makes use of the same set of equations as the online calculation' (line 141) refer to: Does the offline model consider three-dimensional transport (advection and diffusion) of the non-prescribed equations? Which equations exactly are those?
**We thank the reviewer for pointing out that our description of this method was unclear. Upon rereading we realize that it sounds like we have used an offline model, but this is not the case. We have merely performed a simple offline calculation using the output from the NorESM1-ME model. We took the monthly three-dimensional model output (x,y,depth) and put it into Equations 1-3 (in the revised version) to solve for NPP. We assumed a constant euphotic depth of 100m and therefore averaged the inventory over the top 100m for nitrate, phosphate, and dissolved iron to calculate the limiting nutrient in each month. We also used the average temperature in the top 100m and light was attenuated to 50m (in the middle of our depth layer). There are no other equations in our offline calculation than Equations 1-3. The text has been revised to clarify this method.**

Why is the light in the offline calculations attenuated to a constant depth of 50 m, is the offline model two- dimensional or does it resolve depth?

**No, we do not resolve depth. We calculate a value for NPP in the top 100m of the ocean and assume that the light at 50m is a good approximation of average light concentration over the 100m layer.**

One issue that I found particularly confusing in the description of the offline experiments is that N stands for the most-limiting nutrient (phosphate/nitrate/iron). But which nutrient is most limiting is likely to change in the online runs. Are all nutrients prescribed in the offline runs, is there a climatology of the most limiting nutrient?

**In the offline calculation, the most limiting nutrient is computed based on the monthly outputs of nitrate, phosphate, and dissolved iron concentrations. See also our reply above.**

I also have a similar problem with the interpretation of the results of the offline calculations as the first reviewer. The authors use phytoplankton biomass as proxy for assessing the impact of changes in nutrient supply to the euphotic zone due to changes in upper ocean stratification (lines 363-364). What one would really like to use as a control variable in these calculations is the vertical flux of nutrients. I see that nutrient concentrations are probably not a good tracer for this nutrient flux, since they are drawn down to limiting values (assuming sufficient light) regardless of the flux. But the phytoplankton biomass is also just an indirect indicator: Firstly it is also affected by other losses such as zooplankton grazing (as the authors also mention, line 366), to which I would add the sinking losses of biomass through aggregation and sinking: Assume that the only loss of phytoplankton was a quadratic loss through aggregation and sinking. Then biomass would be proportional to the square root of nutrient supply.

**The reviewer is correct and these are very good points. As explained in our**

**reply to reviewer #1 we now calculate a residual term which approximately repre-
sents the integrated circulation-induced changes in phytoplankton and limiting
nutrient. To a first order this term thus includes the advection of nutrients.
The discussion is revised to reflect this. Unfortunately, the vertical fluxes
of nutrients are not available as model outputs. And since the ocean model
is based on isopycnic vertical coordinates, the computation of surface-deep
exchange of nutrients is not straightforward.**

Also, phytoplankton growth rate is affected by both nutrients and temperature, which
however is considered as a separate driver. To me it is thus nor completely clear how
well these two factors can be separated with the offline experiments.
**This point was also raised by reviewer #1. We agree that the presentation of
NPP variation due to changes in phytoplankton was confusing. We now only
compute the total, temperature- and light-induced NPP variability, and discuss
the residual. The residual term predominantly represents the NPP change due
to circulation-induced changes in nutrient and phytoplankton. See also our
reply to reviewer #1.**

A smaller question that I didn't find the answer to in the model description (lines
129-138), and that may affect the interpretation of the manuscript slightly, is whether
the model considers direct effects of ocean acidification (line 536) on carbon cycling
through the marine ecosystem, e.g. by reductions in calcification.
**No. In the HAMOCC model, calcification is indirectly determined by the silicate
availability. In regions of high silicate, biogenic opal production dominates, and
when silicate is low, calcium carbonate production dominates. In the interior
ocean, ocean acidification induced changes in carbonate ion saturation governs
the dissolution rate of calcium carbonate.**

[Figure]

Also, the description of how the different RM methods have been implemented in the model (Lines 163-173) is quite short: to me it was for example a bit unclear how the SAI scenario was modelled. It is said that a layer of sulfate aerosols was prescribed, but then the next sentence states an injection strength, which to me implies that the layer was not prescribed, but calculated as resulting from a balance between injection and some unclear losses.

**The description of the implementation of the RM methods has been clarified. We prescribed a layer in the stratosphere with optical properties representing an injection strength of 20 Tg(S) per year in year 2100, to offset -4.0 W m-2. The aerosol layer was represented by stratospheric zonal aerosol extinction, single scattering albedo and asymmetry factors, as derived from the Tilmes et al. (2015) data set.**

Minor comments

Line 42: At least the CCT method does not act to 'increase the amount of solar radiation reflected' but rather to increase the loss of long-wave radiation passing through the atmosphere.
**This is true, and is the reason for our definition and use of the term Radiation Management (RM) on line 65.**

Line 66 ff: I found this sentence quite confusing: Is it maybe two sentences in one?
**The sentence is revised for clarity and now reads: "As pointed out by Irvine et al. (2016) there are several gaps in the research on the impact of RM on both global climate and the global environment, especially considering that only a few modelling studies to date systematically compare multiple RM methods."**

Line 100: contrary to the statement on line 100 I have not found any presentation of impacts on inorganic carbon in the manuscript, only impacts on air-sea carbon flux. They are of course closely related, but be precise.

**The reviewer are correct that we do not discuss changes in inorganic carbon content by itself, but we do discuss changes in pH as well as air-sea fluxes, which is a part of the inorganic carbon cycle. We agree this was unclear and now state that we look at changes in the inorganic carbon cycle (which is also the title of section 3.2).**

Line 138: It is stated that seawater carbonate chemistry formulation follows the OCMIP protocol. But which one, OCMIP 2 or 3? OCMIP 3 corrected a few smaller errors in the OCMIP 2 protocols.
**The model uses the OCMIP2 protocols and this is now reflected in the text.**

Line 223-225: This result could be emphasised a bit more, it shows why we need full coupled atmosphere-ocean-biogeochemistry models to study this type of effects
**It is indeed important to use full Earth system models to address the climate responses and implications of RM scenarios. The so-called "monsoon-like" response to the tropical and extra-tropical circulation as a result of the MSB forcing has been discussed in several papers before, and we will hence not spend too much time on it in this paper.**

Line 297: 'production' missing after 'increasing primary'
**This is now changed, and primary production is replaced with NPP throughout.**

Line 299-300: 'after termination it takes less than 5 years': What sets the timescale, the atmosphere (radiation), or the ocean biology?
**This timescale is set by the atmosphere. The ocean biology reacts to the (very) fast atmospheric response to termination of RM. We have added a sentence reflecting this.**

Line 327: 'Only CCT significantly changes.': Does that not contradict what has been

said before? Maybe I did not understand what should be said here.
**This section discusses the offline calculations only, the results of which differ somewhat from the model experiment. The sentence is revised to clarify this and now reads: "For the top 100 m of the ocean, the offline calculation shows that only CCT significantly changes NPP compared to RCP8.5."**

Line 336-337: insert 'the' in 'once terminated, CCT method.' **Done**

Line 441: Is 18 percent really a 'minor change' compared to 13 percent?
**Considering the uncertainties in NPP change I'd say these numbers are very similar. However, I agree with the reviewer that the statement may be misleading so "marginally" has been removed.**

Line 447 ff: This and the next paragraph talk about reduction on NPP; it would be clearer if the percent changes would therefore have a negative sign also. **That is true and the paragraphs have been changed accordingly.**

Line 477: 'are quite different': It would be good to have a short summary of the differences, so the reader does not have to read Partanen et al. (2016) herself.
**This is a good suggestion from the reviewer and we have now added a brief description of the major differences between our results and those of Partanen et al (2016), as follows "Overall, the effects of MSB in this study and that of Partanen et al. (2016) are quite different. Spatially, Partanen et al. (2016) sees a very strong correlation between the regions where the MSB forcing was applied and the regions of strongest NPP change which is not apparent in this study. Temporally, the change in NPP in Partanen et al. (2016) comes in form of a relatively rapid decrease over the first ten years MSB is applied while in this study the change is more even throughout the period of MSB forcing."**

[Figure]

Line 563 ff, references: It the Ahlm paper still in the discussion forum or is there a citable full reference by now?

**A revision has been in review and is now accepted with minor revisions.**

---

## Author Response (AR2)

Dear Siv Lauvset,

I have now received the reviewers reports on the revised version of your paper. As you will find, one reviewer has still a small problem with the calculation of the available light for the off-line calculations. It would be ideal if you could rerun the off-line calculations following the reviewers suggestion, and check if/how much this changes your conclusions.

**Reply: Thank you for your comments. I have done the additional calculation the reviewer asks for and find that there is no change (see also my reply to reviewer #2 below). I have added one sentence to this effect (explaining that attenuating light to 50m is equivalent to using an average over the 100m).**

In addition, I have a few additional suggestions:

in Eq. 1, $N_0$ is not defined ln 169 insert 'mean' after 'zonal'

ln 172 emissions 'of'

ln 173 I suggest to put 'accumulation mode' in quotes
or add the diameter in parenthesis as not many readers of BG might be familiar with the term
also: ...emissions...'were' increased ln 174 allowing 'for' the full interactive cycle
(or 'allowing the simulation of' the full...)

ln 200 global mean SST 'is' projected ln 213 oxygen is not a physical variable...
perhaps 'physically driven, such as surface oxygen' is meant?

ln 223 the spatial 'distribution of' the absolute change in SST...

ln 226-228 I'm a bit puzzled how one can see the 'changes in spatial patterns'
in the zonal means figure. Perhaps this can be clarified.

ln 300 All RM 'experiments' (not methods) also exhibit...

ln 314/315 I suggest to change the sentence to
...it takes less than five years for ocean NPP to decrease to RCP8.5 levels.

ln 346 ...upwelling regions (add 's')

ln 690 Journal of 'C'limate

**Reply: All the changes suggested above have been implemented. With the exception of changing**

**"methods" to "experiments" on line 300. "Methods" is the word used throughout the manuscript**
**and we feel that it is more appropriate to keep it consistent.**

Report #1
Anonymous Referee #2

The manuscript has basically taken into account all my criticisms from my first review, so I would argue to publish it. There is only one minor point: For the offline calculations, the authors use the incident light attenuated to a depth of 50m. Unlike the nutrients, or the biomass, which are averaged over the upper 100m for the offline calculations, this is not a true average; but calculating an average would probably be easy: average attenuation could be calculated from
$k = k\_water + k\_chl * Chl$
and then the average light could be calculated from

$$I\_av = 1/100m * Int\_0m^{100m} I(z=0) * exp(-k*z) dz$$

I don't expect this to change the main conclusions crucially, but it might give a somewhat different residual NPP as calculated with I(z=50m), and thus a slightly different conclusion about the role of nutrients. Maybe the authors could quickly check that. I don't think it would take too much time.

Otherwise the paper is fine now and I would recommend publication.

**Reply: The light is attenuated by a different equation in the model so we have used that also in our**
**offline calculations. I have previously looked at results obtained when the light is not attenuated at**
**all, and this has a small effect. Using average light (over the top 100m) is not significantly different**
**from using light attenuated to 100m. The figure below shows the NPP$_{light}$ component for the**
**marine sky brightening experiment with two different light attenuations (time on the x-axis). The**
**other experiments show similar differences.**

[revised manuscript text omitted]